



# Comparison of high-resolution climate reanalysis datasets for hydro-climatic impact studies

Raul R. Wood[1,2,3], Joren Janzing[1,2,3], Amber van Hamel[1,2,3], Jonas Götte[1,2,3], Dominik L. Schumacher[3], and Manuela I. Brunner[1,2,3]

[1]WSL Institute for Snow and Avalanche Research SLF, Davos Dorf, Switzerland
[2]Climate Change, Extremes and Natural Hazards in Alpine Regions Research Center CERC, Davos Dorf, Switzerland
[3]Institute for Atmospheric and Climate Science, ETH Zurich, Zurich, Switzerland

**Correspondence:** Raul R. Wood (raul.wood@slf.ch)

**Abstract.** Continuous high-quality meteorological information is needed to describe and understand extreme hydro-climatic events, such as droughts and floods. Information of highest quality relying on observations is often only available on a national level and for few meteorological variables. As an alternative, large-scale climate reanalysis datasets blending model simulations with observations are often used. However, their performance can be biased due to coarse spatial resolution, model uncertainty, and data assimilation biases. Previous studies on the performance of reanalysis datasets either focused on the global scale, on single variables, or on few aspects of the hydro-climate. Therefore, we here conduct a comprehensive spatio-temporal evaluation of different precipitation, temperature and snowfall metrics for four state-of-the-art reanalysis datasets (ERA5, ERA5-Land, CERRA-Land, and CHELSA-v2.1) over complex terrain. We consider climatologies of mean and extreme climate metrics, daily to inter-annual variability, as well as the consistency in long term trends. Further, we compare the representation of extreme events, namely the intensity and severity of the 2003 and 2018 droughts, as well as the 1999 and 2005 floods in Switzerland. The datasets generally show a satisfactory performance for most of these characteristics, exceptions being the representation of snowfall (solid precipitation) and the number of wet days in ERA5 and ERA5-Land. Our results show clear differences in the representation of precipitation among datasets and a substantial improvement of the representation of precipitation in CERRA-Land compared to the other datasets. In contrast to precipitation, temperature is more comparable among datasets, with CERRA-Land and CHELSA showing smaller biases yet a clear increase of bias with elevation. All datasets are able to identify the 2003 and 2018 drought events, however, ERA5, ERA5-Land, and CHELSA overestimate their intensity and severity, while CERRA-Land underestimates it. The 1999 and 2005 floods are overall well represented by all datasets, with CERRA-Land showing the best agreement with observations and the other datasets overestimating the spatial extent of the events. We conclude that overall, CERRA-Land is the most reliable dataset and suitable for a broad range of analyses, particularly for regions where snow processes are relevant and for applications where the representation of daily to inter-annual precipitation variability is important.



## 1 Introduction

Continuous and high-quality gridded meteorological datasets are crucial to describe, understand and monitor extreme hydro-climatic events, such as droughts and floods. However, identifying suitable meteorological datasets for hydrological applica-
tions is challenging, especially over complex terrain, where climate is influenced by orographic effects (Napoli et al., 2019) and only few observational stations providing high-quality information are available. Many different types of datasets exist ranging from gauge-based data, gridded products interpolated from these station data, to satellite-derived products, reanalysis products, and products that merged information from multiple sources (Roca et al., 2019; Vidal et al., 2009; Soci et al., 2016). Among these, the highest quality datasets are those relying on spatially interpolated observations. Such gridded observation-based
products provide temperature and precipitation fields by spatially interpolating point information from measurement stations onto a grid using different deterministic interpolation schemes of varying complexity (e.g., Hofstra et al., 2008; Daly et al., 2008; Rauthe et al., 2013; Frei, 2013). These datasets are often considered as ground truth or benchmark, especially in regions with dense station networks and long homogeneous records. While of high quality, these observation-based products are often only available on the national level (e.g., Frei, 2013; Krähenmann et al., 2016), or for some trans-boundary catchments (e.g.,
Rauthe et al., 2013; Lussana et al., 2019; Isotta et al., 2013), and most of them are only available for a limited number of meteorological variables (i.e., precipitation and/or temperature). However, consistent data across larger scales and for multiple variables are desirable, as droughts and floods often extend across large regions. Reanalysis datasets, unlike purely observation-based datasets, are available over larger domains and provide physically consistent information on multiple variables including precipitation and temperature (e.g., Hersbach et al., 2020; Gelaro et al., 2017).

Reanalysis datasets are based on numerical models (Dee et al., 2014) with the goal to create a realistic representation of the atmospheric, ocean, and land-surface states of the past. Reanalysis systems include different schemes of data assimilation of various observed surface and atmospheric conditions (e.g., pressure, humidity, temperature and wind) to constrain the simulations to the large-scale observed earth system states (Hersbach et al., 2020; Ridal et al., 2024). For example, in the case of a large-scale drought event caused by a stable high-pressure system, the reanalysis system will simulate a compa-
rable blocking-type weather regime thanks to assimilated pressure and wind fields, but will simulate associated patterns of precipitation anomalies as a response to the modelled system states. Given that for example relevant precipitation processes are unresolved on coarse model scales and hence depend on the internal model structure and implemented physical equations (i.e., parameterization), they will suffer from model uncertainties. Therefore, simulated precipitation and temperature fields can clearly differ from local observations and vary greatly across different reanalysis datasets relying on different modeling
systems (Alexander et al., 2020; Sun et al., 2018).

As it is computationally expensive to run reanalyses for the continental and global scale, large-domain simulations often come at the cost of limited spatial resolution. The current generation of the most widely used global reanalysis datasets has spatial resolutions between 31 and 60 km and includes the fifth-generation European Centre for Medium-Range Weather Forecasts (ECMWF) reanalysis dataset (ERA5, ≈31 km; Hersbach et al., 2020), the Modern-Era Retrospective analysis for
Research and Applications, Version 2 (MERRA-2, ≈50 km; Gelaro et al., 2017), and the Japanese 55-year Reanalysis (JRA-55,



≈79 km; Kobayashi et al., 2015). Because of their coarse spatial resolution, these datasets fail to deliver information at impact-relevant scales. To provide higher-resolution climate information and to better represent water and energy cycles over land, the ECMWF has produced an offline land-surface model simulation forced with bilinearly interpolated atmospheric fields from ERA5, resulting in the ERA5-Land dataset at ≈ 9 km spatial resolution (Muñoz-Sabater et al., 2021). At continental
scales, even higher-resolution regional reanalysis products can be produced by dynamically downscaling global reanalysis information over a limited area with or without an additional data assimilation scheme (Bollmeyer et al., 2014). Thereby, the regional reanalysis system is one-way nested within the global reanalysis, which means that the global reanalysis delivers the large-scale boundary and initial conditions for the higher resolution numerical model over the target region. Examples for such regional reanalysis products include the Copernicus European Regional Reanalysis (CERRA) at 5.5 km horizontal resolution
(Ridal et al., 2024) and the 6 km COSMO-REA6 from the German Weather Service (Bollmeyer et al., 2014). Because many applications require even higher resolution data, there are many efforts to statistically downscale global reanalysis datasets in a physically meaningful way, for example, the Climatologies at High resolution for the Earth's Land Surface Areas (CHELSA) datasets (Karger et al., 2017, 2021b), which provide refined information at approx. 1 km resolution.

The main advantage of global and regional climate reanalyses is that they provide continuous and physically consistent time
series of many surface and atmospheric variables across space and time for the past. Therefore, they are increasingly used to study and model hydro-climatic extremes. The choice of a suitable dataset for a particular hydrological application is not straightforward, owing to the wide range of available data products (by various provider) at different temporal and spatial resolutions, with different spatial domains. Such a choice requires information on dataset performance with respect to key climatic variables, including temperature and precipitation. In other words, thorough comparisons between reanalysis datasets
and gridded observations are required, as both are used as climatic reference conditions in many different applications, including understanding the drivers of drought (e.g., Bakke et al., 2020; Brunner et al., 2023); tracking the propagation of drought from the atmosphere to the hydrosphere (Brunner and Chartier-Rescan, 2024); simulating streamflow with hydrological models to understand the risk of floods (e.g., Brunner and Fischer, 2022; Willkofer et al., 2020); analyzing elevation-dependent trends in mean and extreme precipitation across mountain regions (Ferguglia et al., 2024); evaluating climate model performance and
bias adjustment (e.g., Vautard et al., 2021; Tootoonchi et al., 2022).

There is ample evidence that the choice of a reanalysis dataset can influence the results and conclusions of hydrological impact studies (e.g., Kotlarski et al., 2017; Gampe et al., 2019; Tarek et al., 2021). Therefore, several previous studies have compared different sets of reanalysis datasets with respect to different climate characteristics. For example, Bandhauer et al. (2021) have evaluated ERA5 precipitation data in three mountain regions in Europe compared to high-resolution gridded
observations, and Lavers et al. (2022) for extreme precipitation compared to 5,637 precipitation stations quasi-globally; Gebrechorkos et al. (2024) assessed the value of ERA5-Land precipitation for streamflow simulations worldwide and Tarek et al. (2020) for ERA5 in North America; McClean et al. (2023) evaluated the capabilities of global reanalysis products for flood risk modelling in river catchments in Northern England; Dura et al. (2024) analyzed seven gridded precipitation products, among them ERA5-Land and CERRA-Land, for their suitability to estimate precipitation enhancement with altitude in France;
and Monteiro and Morin (2023) compared ERA5, ERA5-Land and CERRA-Land, among other datasets, with respect to snow





depth and snow cover duration in the European Alps. Most of these existing studies evaluating reanalysis datasets focused on one specific aspect, for example the representation of precipitation. In addition, they stratified their analyses into very large spatial units, for example the entire globe or mesoscale catchments and regions, which can hide dataset differences. Further, only few of these studies provide information on reanalysis dataset performance with respect to the representation of extreme events, daily to interannual variability, and temporal trends. Therefore, while all of these studies provide valuable insights with respect to certain climatic characteristics for specific regions, it remains unclear how well different datasets perform in mountain regions in terms of temperature and precipitation characteristics. However, a good performance in mountain regions is essential for climate monitoring as they are identified as hotspots of climate and hydrological trends (Adler et al., 2022). A thorough reanalysis dataset comparison is needed in these regions because dataset performance may vary greatly depending on elevation (Monteiro and Morin, 2023; Dura et al., 2024). Furthermore, an accurate representation of both temperature and precipitation as well as their interplay is crucial in these regions because it determines the partitioning of liquid and solid precipitation, and hence the build-up of snow storage. Along the climate-hydrological modeling chain, a misrepresentation of either of these variables can lead to a misrepresentation of hydrological extremes including floods and droughts. Therefore, the representation of the all of these components is crucial for many hydro-climatic impact studies in particular in regions with complex topography.

To shed light on the question which reanalysis products are most suitable for hydrological impact studies in the mountain regions of Europe, we compare four state-of-the-art and widely used global and regional reanalysis datasets with gridded observations with respect to different climate variables that are crucial to describe hydrological behavior, namely temperature and precipitation. We focus our analysis on Switzerland because it shows a large climatic gradient owing to its complex topography, and because high-quality observation-based gridded datasets are available for benchmarking. In order to account for the variety in available reanalysis datasets, we compare the global ERA5 reanalysis dataset, its higher-resolution version over land (ERA5-Land), the regional reanalysis dataset for Europe (CERRA), and a statistically downscaled global reanalysis dataset (CHELSA). We evaluate these datasets by quantifying differences between model simulations and gridded observations for different climate metrics including mean and extreme climate metrics (Sect. 4.1, 4.2); precipitation and temperature variability across various temporal scales (Sect. 5.1); the consistency in long-term trends (Sect. 5.2); snowfall estimates as a result of precipitation-temperature dependence (Sect. 4.3); and the spatial and temporal representation of observed severe droughts (Sect. 6.1), and heavy precipitation events which led to flooding in Switzerland (Sect. 6.2).

## 2 Datasets

For our reanalysis dataset comparison, we use temperature and precipitation data from four reanalysis products as well as two gridded observational datasets as our benchmark (Table 1). We limit our comparison to the period 1986–2020, based on the dataset with the shortest temporal coverage, i.e. the Copernicus European Regional ReAnalysis (CERRA). Although some reanalysis products provide data at sub-daily time resolution, the comparison is performed at a daily resolution, which is the



**Table 1.** Overview of the datasets used in this study.

| Dataset | Type | Spatial Resolution | Spatial Coverage | Temporal Coverage | Reference |
|---------|------|--------------------|------------------|-------------------|-----------|
| ERA5 | Reanalysis | 31 km | Global | 1950 - present | Hersbach et al. (2020) |
| ERA5-Land | Reanalysis | 9 km | Global | 1950 - present | Muñoz-Sabater et al. (2021) |
| CERRA(-Land) | Reanalysis | 5.5 km | Europe | mid 1984 - mid 2021 | (Schimanke et al., 2021; Verrelle et al., 2022) |
| CHELSA | Downscaled reanalysis | 30 arcsec (≈ 1 km) | Global | 1979-present | Karger et al. (2023) |
| TabsD / RhiresD | Gridded observations | 75 arcsec (≈ 2 km) | (Hydrological) Switzerland | 1961-present | (MeteoSchweiz, 2021a, b) |
| CAMELS-CH | Catchment outlines and elevations | | Hydrological Switzerland | | Höge et al. (2023b) |

resolution of the observational benchmark dataset. As the datasets provide climate information at various grid resolutions and grid specifications, we quantify dataset differences based on catchment averages (Figure 1).

## 2.1 Gridded observations

We use two gridded observational products from the Swiss Federal Office of Meteorology and Climatology (MeteoSwiss) as our benchmark products. These products use daily observations from measurement stations, which are spatially interpolated onto a 75 arcsec grid (≈ 2 km) over Switzerland. We use daily mean temperature (*TabsD*), which is based on data from 90 high-quality stations (MeteoSchweiz, 2021b; Frei, 2013) and precipitation (*RhiresD*), which provides total precipitation over a day starting from 06:00 UTC and is based on data from 650 measurement stations (MeteoSchweiz, 2021a; Schwarb, 2000). Here, these gridded MeteoSwiss products serve as a purely observation-based reference for the comparison of reanalysis datasets.



## 2.2 Reanalysis datasets

### 2.2.1 ERA5 and ERA5-Land

We use two reanalysis datasets from the ERA5 product family, namely, ERA5 and ERA5-Land. ERA5 is a global reanalysis dataset produced by the European Centre for Medium-Range Weather Forecasts (ECMWF) with a spatial resolution of 31 km and hourly temporal resolution (Hersbach et al., 2020). It is the coarsest resolution dataset that we consider. ERA5 is created with the Integrated Forecasting System Cy41r2 and covers the period from 1940 until the present. ERA5-Land uses linearly interpolated atmospheric forcing from ERA5 data to run the CHTESSEL (Carbon Hydrology-Tiled ECMWF Scheme for Surface

Exchanges over Land) land surface model at a spatial resolution of 9 km (Muñoz-Sabater et al., 2021). The dataset provides improved and additional land surface variables (e.g., soil moisture, snow, or hydrological variables) compared to ERA5. The ERA5-Land precipitation data are provided as linearly interpolated fields from ERA5 without any adjustments – by linear interpolation onto the ERA5-Land grid (Muñoz-Sabater et al., 2021)– and accumulated to 24 hours, whereas ERA5-Land temperature data were interpolated and adjusted by daily lapse rates to account for the altitude differences between the ERA5 and

ERA5-Land grids (Dutra et al., 2020; Muñoz-Sabater et al., 2021), and are provided at an hourly time step.

### 2.2.2 CERRA

In addition to the two global reanalyses products, we use the Copernicus European Regional ReAnalysis (CERRA), which is provided for Europe for the period from mid-1984 until mid-2021. The CERRA reanalysis system comprises multiple datasets,

from which we use the CERRA high-resolution dataset (Schimanke et al., 2021; Ridal et al., 2024) at 5.5 km resolution and 3-hourly temporal resolution for temperature, and the CERRA-Land dataset (Verrelle et al., 2022) for precipitation at 5.5 km resolution and daily accumulation. CERRA is a classic reanalysis system based on the HARMONIE-ALADIN model (Bengtsson et al., 2017; Termonia et al., 2018), which utilizes data assimilation for the atmosphere and surface. Since it is a regional reanalysis system, it requires lateral boundary conditions, which are based on ERA5 model simulations (Ridal et al., 2024).

CERRA-Land is a standalone simulation of the land surface model SURFEX v8.1 which delivers additional land surface variables. SURFEX is driven by atmospheric variables from the CERRA high-resolution simulations and uses precipitation from the regional precipitation analysis system MESCAN (Soci et al., 2016; Ridal et al., 2024). The MESCAN regional precipitation analysis system uses precipitation fields from the CERRA simulations as a first guess and incorporates additional in-situ observational rain gauge data through optimal interpolation (Soci et al., 2016). Because of this additional precipitation data

assimilation, we use the CERRA-Land precipitation product instead of the CERRA product. We will refer to this dataset composed of CERRA temperature and CERRA-Land precipitation as CERRA.



### 2.2.3 CHELSA

The Climatologies at High resolution for the Earth's Land Surface Areas (CHELSA) products are statistically downscaled ver-
165 sions of large-scale reanalysis datasets to a high resolution 30 arcsec grid ($\approx$ 1km at the equator) (Karger et al., 2017, 2021b, 2023).
Here, we use precipitation and temperature data from CHELSA version 2.1 (Karger et al., 2023, 2021a), which is based on a
statistical downscaling of the W5E5 global reanalysis dataset (original spatial resolution of 0.5 degrees) (Cucchi et al., 2020).
W5E5 is a bias-adjusted version of ERA5 over land. The CHELSA high-resolution temperature data are based on an atmo-
spheric lapse-rate downscaling (Karger et al., 2017), considering differences in the orography between the 30 arcsec CHELSA
170 topography –based on the global multi-resolution terrain elevation data GMTED2010 (Danielson and Gesch, 2011)– and the
W5E5 topography, and lapse rates determined for different atmospheric pressure levels in ERA5. The precipitation data are
based on a downscaling algorithm which uses spatial wind fields and boundary layer thickness to account for orographic wind
effects (Karger et al., 2021b). The precipitation fluxes are thereby preserved at the 0.5° grid of the parent W5E5 dataset.





**Figure 1.** Mean annual climatology for 1986–2020 for the five datasets at their original grid resolution. (A) Mean annual precipitation for gridded observations, ERA5 reanalysis, ERA5-Land reanalysis, CERRA reanalysis, and CHELSA reanalysis (upper left to bottom right panels). (B) Mean annual temperature for the five datasets. Plots are overlaid with the outline of the 97 headwater catchments of the CAMELS-CH dataset (solid) and the Swiss country borders (dotted). The observations in (A) and (B) are further overlaid with the catchment outlets (solid dots).



## 2.3   Catchments CAMELS-CH

For the calculation of catchment averages, we rely on catchment delineations from the CAMELS-CH dataset (Catchment Attributes and MEteorology for large-sample Studies - Switzerland; Höge et al. (2023b)). CAMELS-CH is a large sample hydro-meteorological dataset providing catchment outlines and static attributes for 331 catchments in Switzerland and neighboring countries (i.e., hydrological Switzerland). We limit the analysis to 97 non-overlapping headwater catchments within the political borders of Switzerland. While all reanalysis datasets have data beyond the borders of Switzerland, the gridded temperature observations are limited to political Switzerland. For the analysis of elevation dependence, we use catchment elevation from the CAMELS-CH dataset.

## 3   Methods

To describe and identify the most important differences between datasets, we compare the four reanalysis datasets to the gridded observations for a broad range of precipitation, temperature, and snowfall metrics calculated (Sect. 3.1) at the catchment level. First, our comparison focuses on both absolute and relative differences between the climate metrics derived from the reanalysis datasets and those derived from the gridded observations. Second, we use these climate metrics to analyze the temporal consistency of precipitation and temperature variability (Sect. 3.2.1) as well as the consistency in long-term trends (Sect. 3.2.2). Last, we use catchment time series of precipitation to compare the spatial and temporal representation of two severe droughts (2003 and 2018), and two heavy precipitation events, which led to severe floods in Switzerland (1999 and 2005) (Sect. 3.3).

To calculate catchment averages, we use two complementary approaches: a time series-based approach, which first calculates time series of daily catchment averages before calculating the metric of interest; and an metric-based approach, which first calculates the metric of interest on the native dataset grid before averaging the metric over the catchment. The time series-based perspective provides information about the average state of the catchment from a hydrology perspective, while the metric-based perspective allows us to compare the average behavior of the index at the catchment scale.

### 3.1   Climate metrics

First, we compare general climatic characteristics, such as long-term mean daily precipitation and temperature at annual and seasonal scales, which are based on catchment-average time series of daily precipitation and temperature. Dataset differences are expressed as absolute and relative differences between metrics derived from the reanalysis datasets and those derived from observations. Further, we check whether these differences are elevation dependent, i.e. vary with catchment elevation.

Second, we calculate a selection of univariate annual extreme and non-extreme precipitation and temperature metrics. To describe non-extreme precipitation characteristics, we consider the annual number of wet (*wetdays*) and dry days (*drydays*), and the maximum number of consecutive wet (*cwd*) and dry days (*cdd*) within a year. To describe extreme precipitation characteristics, we use the annual maximum accumulated precipitation over 1-day (*Rx1d*), 2-days (*Rx2d*), and 5-days (*Rx5d*),



and the fraction of total annual accumulated precipitation falling on wet days (*R95pTot*) and on very wet days (*R99pTot*). To describe temperature, we consider the annual number of cold days (*colddays*), the annual maximum of daily mean temperature (*tg_max*), and the annual minimum of daily mean temperature *(tg_min).*

Last, we compare the datasets with respect to their representation of snowfall (i.e., solid precipitation), which is influenced by the interdependence of precipitation and temperature. While ERA5, ERA5-Land and CERRA explicitly represent snow, CHELSA and observations do not. Therefore, we consistently approximate the snowfall using a common temperature threshold for all reanalysis datasets and the observations. We separate precipitation into liquid and solid precipitation using 0°C as a threshold, with *liquid precipitation* and *solid precipitation* falling above and below 0°C, respectively. Our analysis focuses on the representation of total accumulated solid precipitation (*solidprcptot*) and the fraction of solid to total accumulated precipitation (liquid+solid) – both computed for the hydrological year (October-September).

A complete list of the metrics used for the comparison and their definitions are provided in Table 2. All climate metrics were calculated with the *xclim python package* (Bourgault et al., 2023b).

## 3.2 Temporal consistency

### 3.2.1 Daily to annual variability

To compare the temporal consistency, we analyze the representation of daily to inter-annual temperature and precipitation variability. Temperature variability is expressed as the standard deviation of temperature at different time scales, e.g., the standard deviation of daily means (i.e. daily variability) or the standard deviation over annual means (i.e., inter-annual variability). Precipitation variability is expressed as the coefficient of variation, which is defined as the standard deviation divided by the mean. For temperature, we opted against using the coefficient of variation due to mean values being close to zero, and use standard deviation instead. We compare the representation of inter-annual variability for annual and seasonal mean precipitation and temperature. Seasonal variability is defined as the year-to-year variability between the same season, e.g., winter-to-winter variability (DJF). Further, we analyze the inter-annual variability of the maximum 1-day and 5-day precipitation and the minimum and maximum of daily mean temperature.

### 3.2.2 Trends

Next, we analyze the consistency of the presence of significant long-term trends and their trend magnitudes in various metrics across the different datasets and assess how well trends in the reanalysis match observed trends. For trend significance, we apply the Mann-Kendall test (Mann, 1945; Kendall, 1975) using 0.05 as the significance level. Further, as trends in precipitation are typically masked by large internal variability (Wood and Ludwig, 2020; Wood, 2023), we additionally consider trends as weakly significant if the p-value lies between 0.05 and 0.1. Trends with p-values above 0.1 are considered non-significant and are labeled as no trend. The trend magnitude and sign are estimated using the Theil-Sen slope estimator (Sen, 1968). To assess the spatial consistency of trends, we compare the trends of each reanalysis dataset with the trends in observations and test for



**Table 2.** Definition of precipitation and temperature metrics

|  | Acronym | Metric name | Definition | Unit |
|---|---|---|---|---|
| Precipitation (univariate) | prcptot | Total accumulated precipitation | Total accumulated precipitation (liquid & solid) | mm |
|  | Rx1d | Maximum 1-day precipitation | Annual maximum 1-day accumulated precipitation amount | mm |
|  | Rx2d | Maximum 2-day precipitation | Annual maximum 2-day accumulated precipitation amount | mm |
|  | Rx5d | Maximum 5-day precipitation | Annual maximum 5-day accumulated precipitation amount | mm |
|  | R95pTot | Fraction of precipitation due to very wet days | Fraction of total annual precipitation amount due to wet days with daily precipitation >95th percentile | % |
|  | R99pTot | Fraction of precipitation due to extremely wet days | Fraction of total annual precipitation amount due to wet days with daily precipitation >99th percentile | % |
|  | wetdays | Number of wet days | Number of wet days per year with daily precipitation $\geq$ 1mm/d | days |
|  | drydays | Number of dry days | Number of dry days per year with daily precipitation < 1mm/d | days |
|  | cwd | Maximum number of consecutive wet days | Maximum number of consecutive days with daily precipitation $\geq$ 1mm/d | days |
|  | cdd | Maximum number of consecutive dry days | Maximum number of consecutive days with daily precipitation < 1mm/d | days |
| Precipitation (multivariate) | solidprcptot | Total accumulated solid precipitation | Total accumulated solid precipitation approximated by precipitation on days with daily mean temperature below 0°C | mm |
|  | liquidprcptot | Total accumulated liquid precipitation | Total accumulated liquid precipitation approximated by precipitation on days with daily mean temperature above 0°C | mm |
|  | solidprcpratio | Fraction of solid precipitation to total precipitation | The fraction of solidprcptot to prcptot. | % |
| Temperature (univariate) | tg-max | Maximum of daily mean temperature | Annual maximum of daily mean temperature | °C |
|  | tg-min | Minimum of daily mean temperature | Annual minimum of daily mean temperature | °C |
|  | colddays | Number of cold days | Number of days with daily mean temperature <25th percentile | days |



each catchment whether the two datasets agree on the significance and sign of the trend. We label the test as true when both datasets show a significant trend and the same sign (*true: trend*) or when both datasets show no significant trend (*true: no trend*). The test is labeled false when either the observations show a significant trend and the reanalysis dataset shows no trend (*false: undetected trend*) or when both datasets show a significant trend but they don't agree on the sign (*false: trend*). If the reanalysis dataset shows a significant trend in the absence of a significant trend in the observations, we also label this mismatch as false (*false: trend*).

## 3.3 Extreme event analysis

Last, we compare the reanalysis datasets with respect to their ability to represent observed extreme events. Specifically, we analyze the consistency among the datasets for two distinct extreme event types, meteorological droughts and extreme precipitation. We compare the spatial and temporal representation of the droughts in 2003 and 2018 as well as the extreme precipitation events that triggered floods in 1999 and 2005 in Switzerland. For both event types, we compare the severity — expressed by standardized precipitation values — and the intensity of the events, expressed by cumulative 6-month precipitation deficits (for droughts) or 2-day precipitation sums (for extreme precipitation).

### 3.3.1 Meteorological drought

In the summers of 2003 and 2018, Switzerland was affected by severe drought conditions (Brunner et al., 2019). In 2003, the drought resulted in largely reduced streamflow in the Rhine (-46% of normal summer flow) and in the Aare catchments (-38%) (Zappa and Kan, 2007). In 2018, Switzerland on average only received 57% of its normal precipitation amount between April and September (MeteoSchweiz, 2019).

We analyze differences in the drought intensity between datasets by comparing the cumulative precipitation deficits from March until August in 2003 (2018). The severity of the events is compared based on the widely used Standardized Precipitation Index (SPI) for a 6-month accumulation period (March-August), which we computed by transforming the 6-month sums to the standard normal distribution using the Gamma distribution (Lloyd-Hughes and Saunders, 2002; Stagge et al., 2015).

### 3.3.2 Extreme precipitation

In May 1999 and August 2005, Switzerland was affected by multiple extreme precipitation events, which triggered sever flooding in different parts of Switzerland. For the May 1999 flood, we compare two precipitation events which occurred within 10 days of each other (11–12 May and 21–22 May) and caused flooding in the Swiss Midlands (Hilker et al., 2009). For the May 2005 flood, we compare the single two-day precipitation event on 21–22 May, which affected the northern slopes of the Swiss Alps and caused widespread flooding in Central Switzerland and the Bernese Oberland (Beniston, 2006; Hilker et al., 2009). Similar to the drought analysis, we compare the representation of these events in terms of their intensity and severity. The intensity is defined as the 2-day precipitation sum during each of the events and the severity is described using standardized precipitation. In contrast to the SPI-6 calculation for droughts, we apply standardization to all 2-day rolling precipitation sums



in May or August. In order to also include extreme events from the adjacent months, we include +/- 15 days of the previous/past
month to the rolling window calculation. Here, we fitted a generalized extreme value distribution prior to the transformation
to a standard normal distribution to retrieve the 2-day SPI values. Precipitation sums less then 1 mm were excluded from the
standardization to reduce the influence of the large number of zero precipitation days.

## 4   Mean and extreme precipitation and temperature climatology

### 4.1   Mean climatology

The four reanalysis datasets differ only slightly in terms of their annual and seasonal precipitation and temperature climatology
and some biases exist with respect to the observations (Figure 2). Simulated mean daily precipitation is slightly overestimated
across catchments by most reanalysis datasets — except for CERRA — both at an annual and seasonal time scale (Figure 2a).
In summer, all reanalysis datasets overestimate precipitation with respect to observations. The positive reanalysis biases are
particularly evident in catchments at high- and low-elevations (>2000 and < 1000 m.a.s.l.), and less pronounced in catchments
at mid-elevations (1000-2000 m.a.s.l.) for all reanalysis datasets (Figure S1). In contrast, simulated mean daily temperatures
match observations well on average across all catchments for all datasets at an annual scale and for most seasons, an exception
being winter when three out of four reanalyses datasets slightly underestimate temperature. While reanalysis biases in temper-
ature are small over all catchments on average, they vary substantially across catchments, are more often negative than positive,
and show some dependence on elevation for CERRA and CHELSA, which under- and overestimate temperature at high- and
low-elevations, respectively (Figure S2). ERA5 shows neither clear spatial patterns in the biases nor a clear elevational depen-
dence of the biases. However, it shows generally more overestimation than underestimation across catchments, compared to
ERA5-Land, which generally shows an underestimation.





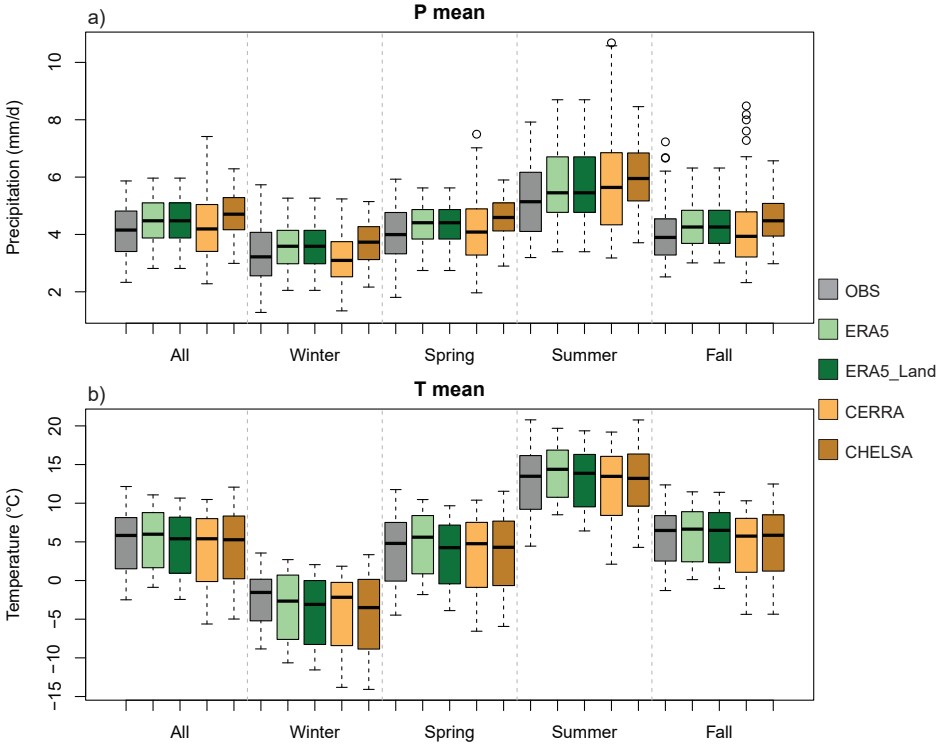

**Figure 2.** Comparison of the mean daily catchment (a) precipitation and (b) temperature climatology of the four reanalysis datasets (ERA5, ERA5-Land, CERRA, and CHELSA) with the gridded observations of MeteoSwiss for the entire year (all) and the four meteorological seasons (winter: Dec–Feb, spring: Mar–May, summer: Jun–Aug, fall: Sep–Nov) across all catchments.

## 4.2 Extreme climatology

While reanalysis datasets only slightly vary in terms of mean climatology, they can substantially differ for certain extreme

metrics, meaning that some products show stronger biases compared with observations than others (Figure 3). These biases are strongest for extreme precipitation metrics, namely, mean annual maximum 1-day precipitation (rx1day; Figure 3a), fraction of total precipitation related to very wet days (r99ptot; Figure 3b), and mean number of wet days per year (wetdays; Figure 3c). All of these metrics are on average under- or overestimated by all reanalysis products, except by CERRA, across all catchments. Rx1day and r99ptot are underestimated by ERA5, ERA5-Land, and CHELSA across all elevation zones (Figure

S3 and S4) as precipitation is distributed across too many wet days in all of these products, especially in catchments at higher elevations (Figure S5). In contrast, CERRA captures both precipitation intensity (rx1day and r99ptot) and the number of wet days well on average across all catchments, with slightly higher dry biases but too many wet days in high-elevation catchments (Figures S3, S4, and S5). Biases in extreme temperature indicators are much less pronounced across all catchments than those related to extreme precipitation metrics (Figure 3e–h). However, they are more often negative than positive for maximum

and minimum temperature (for ERA5-Land, CERRA, and CHELSA) and more often positive than negative for the number





of cold days (for ERA5-Land, CERRA, and CHELSA). In addition, they show a clear relationship with elevation for some reanalysis products. Mean annual maximum daily temperatures tend to be over- and underestimated at high elevations by ERA5 and CERRA, respectively, while the biases of the other two datasets are less related to elevation (Figure S6). Mean annual minimum daily temperatures are underestimated at high elevations by all data products (Figure S7) and the number of
cold days is underestimated by ERA5 and overestimated by CERRA and CHELSA in high-elevation catchments (Figure S8).

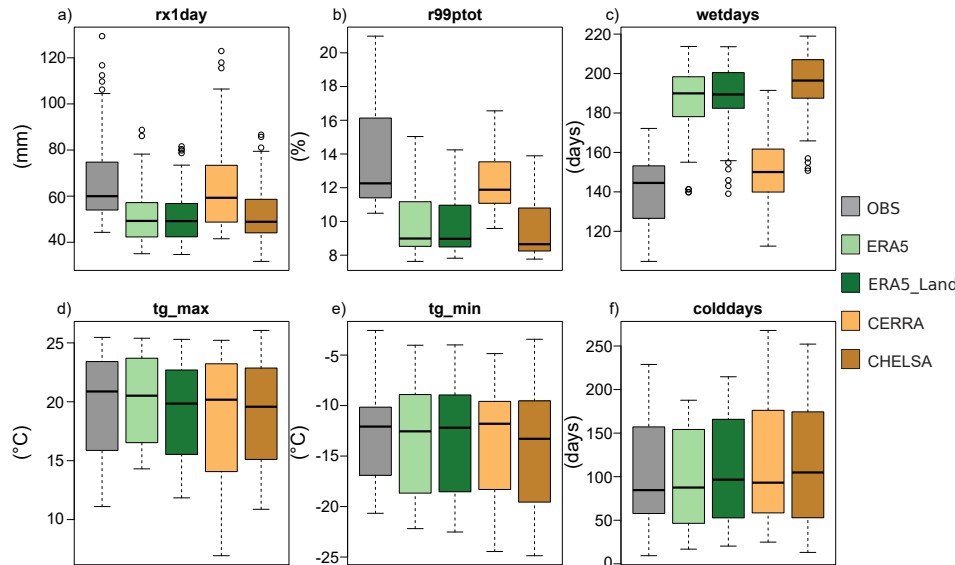

**Figure 3.** Comparison of different (a–c) precipitation and (d–f) temperature metrics derived from the four reanalysis datasets with metrics derived from gridded observations across all catchments: (a) Mean annual maximum 1-day precipitation (rx1day), (b) Fraction of total precipitation related to very wet days (r99ptot), (c) Mean number of wet days per year, (d) Mean annual maximum of daily mean temperature (tg_max), (e) Mean annual minimum of daily mean temperature (tg_min), (f) Mean number of cold days per year.

## 4.3  Solid precipitation

The fraction of solid to total precipitation and the total amount of solid precipitation are best represented by CERRA and CHELSA, which show high agreement with observations, especially in catchments below 1500 m. Thereby, the biases in CERRA seem to be catchment-specific, while CHELSA shows a slight overestimation of solid precipitation with elevation.
ERA5 generally underestimates both the fraction and total amount of snowfall with a clear increase in biases with elevation, while ERA5-Land clearly overestimates solid precipitation at all elevations.

The estimated fraction of solid to total (liquid + solid) precipitation from observations increases linearly from up to 10% in the low-elevation catchments ($\leq$ 1000 m.a.s.l) to 40% and more in the high-elevation catchments ($\geq$ 2000 m.a.s.l) and up to 70% in the highest catchments above 2500 m (Figure 4e). Analogously, the total amount of solid precipitation increases from
on average of less than 100 mm per year in the low-elevation catchments to above 1250 mm in the high-elevation catchments



(Figure 4e). ERA5 generally underestimates the fraction of solid precipitation and shows a clear increase in the bias from lower biases at lower elevation (up to 10 percentage points) to larger biases of up to 20 percentage points at higher elevations (Figure 4a). Such elevation dependence — although reversed — is also apparent when looking at the relative biases in the mean annual amount of solid precipitation with a clear underestimation of up to 100% in the low-elevation catchments and up to 50% in

the middle- to high-elevation catchments (Figure 4a, coloring of dots). In contrast, ERA5-Land overestimates the fraction of solid precipitation by 10-20 percentage points with no clear elevation bias (Figure 4b). As a result, the total amount of solid precipitation is largely overestimated by ERA5-Land, especially at lower elevations (>100% dark purple) compared to higher elevations (25-75%, blues). This difference between ERA5 and ERA5-Land is also apparent when looking at the number of cold days, which is overestimated by ERA5-Land but not by ERA5 (Figure 3). CERRA and CHELSA show small biases for

the estimated fraction of solid precipitation compared to the estimated fraction in observations, especially at low elevations. Both datasets show slightly larger biases with increasing elevation, but remain well below the biases of ERA5 and ERA5-Land. While CERRA shows no clear elevation dependence for an over- or underestimation of the fraction of solid precipitation –differences rather seem to be catchment dependent–, CHELSA shows slightly larger biases in catchments above 1500 m. The biases in the total amount of solid precipitation are comparably low in CERRA and vary between over- and underestimation

depending on the catchment. CHELSA also shows good agreement with respect to the fraction of solid precipitation with very low differences at lower elevations and slightly larger biases at higher elevations. CHELSA generally overestimates the total amount of precipitation by up to 75%.



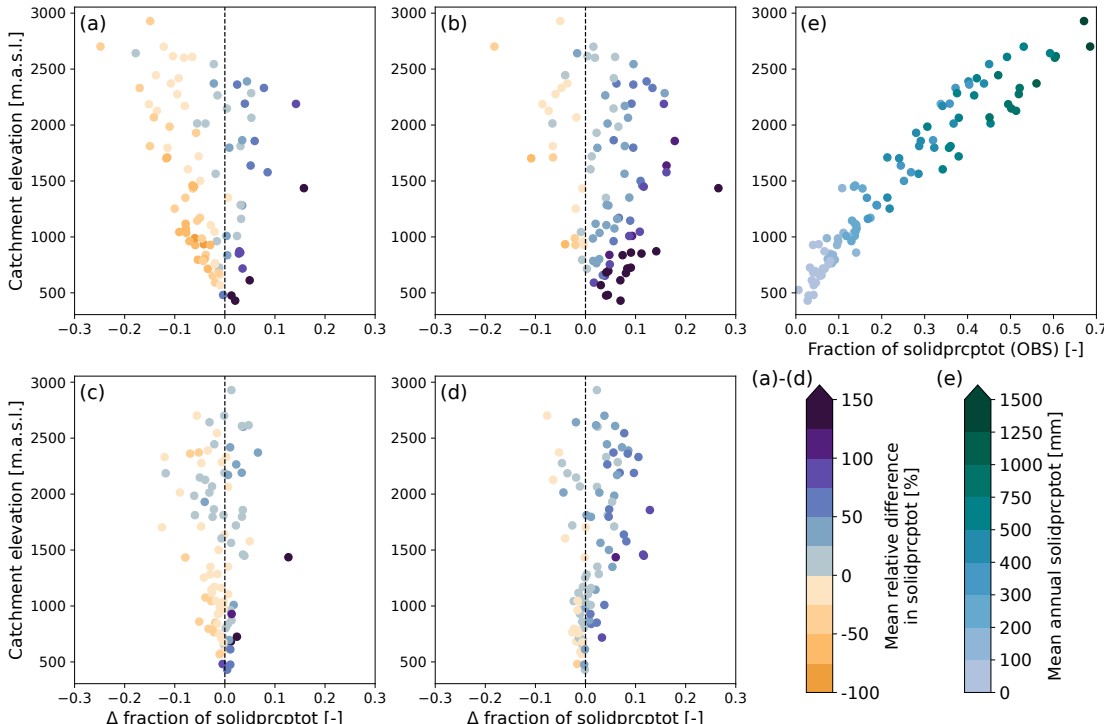

**Figure 4.** Absolute differences in the fraction of solid (solidprcptot) to total precipitation (prcptot) (x-axes of (a)-(d)) and mean relative difference in solidprcptot (coloring of dots in (a)-(d)) for the four reanalysis datasets [(a) ERA5, (b) ERA5-Land, (c) CERRA, (d) CHELSA] compared to observations. Differences are plotted against the respective catchment mean elevation on the y-axis. (e) Shows the fraction of solidprcpt to prcptot of the observations on the x-axis and the mean annual solidprcptot as dot colors. Solid precipitation is estimated in all reanalysis datasets and observations by the same temperature threshold.

## 5 Temporal consistency

### 5.1 Precipitation and temperature variability

The representation of precipitation variability is clearly best in CERRA across all timescales and seasons, while all other datasets (ERA5, ERA5-Land, and CHELSA) underestimate precipitation variability (Figure 5). Daily precipitation variability (i.e., variability from one day to the other) is underestimated by all reanalysis datasets. While ERA5, ERA5-Land, and CHELSA clearly underestimate daily variability, CERRA is closer to observations. The observations show a considerably higher spread of variability across catchments than all of the reanalysis datasets. For monthly variability, CERRA shows very

similar median variability across catchments compared to observations (Figure 5b), but again too little catchment spread in variability. The other datasets clearly underestimate variability as well as catchment spread in variability.



At the inter-annual timescale (Figure 5c), ERA5, ERA5-Land and CHELSA continue to underestimate precipitation variability and CERRA is again closest to observations. If we look at the year-to-year variability in the different seasons (Figure 5d-g), all datasets agree on larger year-to-year variability in winter (DJF) and fall (SON), as well as lower variability in spring 345 (MAM), and show the lowest precipitation variability in summer (JJA). However, while CERRA is in good agreement with observations, the other datasets (ERA5, ERA5-Land, CHELSA) underestimate inter-annual precipitation variability at both annual and seasonal scales. However, all datasets agree on a decreasing level of variability from daily to monthly, and to annual scales.

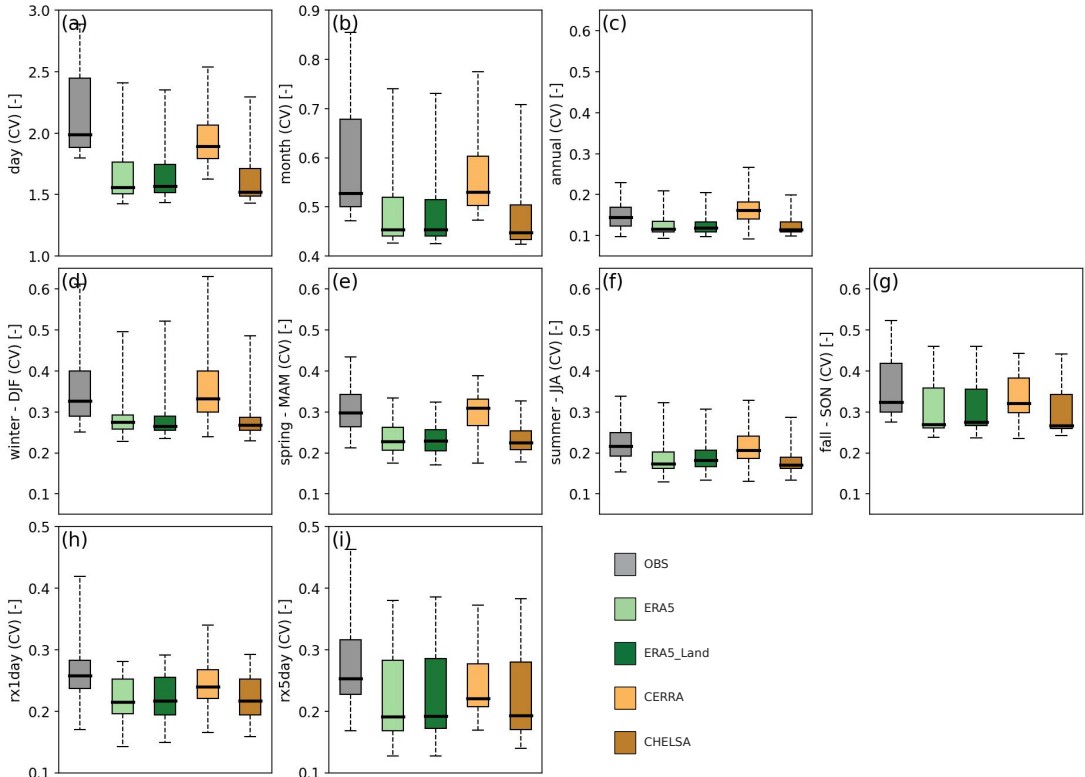

**Figure 5.** Comparison of daily to inter-annual precipitation variability. Boxplots with (a) daily, (b) monthly, and (c) inter-annual precipitation variability. (d)-(g) Show inter-annual variability for different seasons [(d) DJF, (e) MAM, (f) JJA, (g) SON]. (h) Shows the inter-annual variability of rx1day and (i) for rx5day. The boxplots represent variability across catchments and the colors indicate the dataset. Panels (a)-(c) show a different scaling on the y-axis due to the different aggregation periods. Precipitation variability is expressed as the coefficient of variation (CV; Standard deviation divided by the mean).

In contrast to precipitation variability, all datasets are more consistent with observations regarding temperature variability. 350 CERRA is showing an improved representation of temperature variability at daily and monthly scales compared to the other datasets (Figure 6a–b). The inter-annual temperature variability is overestimated by ERA5 and CHELSA, underestimated by



ERA5-Land, and is comparable to observations for CERRA —but with exaggerated inter-annual variability spread between catchments (Figure 6c). All reanalysis datasets agree with observations on the seasonal course of variability and show the largest temperature variability in winter and considerably lower variability in the other seasons (Figure 6d–g). In winter, all
datasets underestimate temperature variability, with ERA5 and ERA5-Land showing the weakest and strongest underestimation, respectively. In spring, ERA5-Land also clearly underestimates year-to-year variability, while all other datasets are comparable with observations. In summer and fall, all datasets are more or less consistent with observations. Looking at the inter-annual variability of the maximum (minimum) of daily mean temperature, all datasets underestimate the variability of the temperature on the warmest and in particular the coldest day of the year (Figure 6h,i).

In summary, CERRA best represents precipitation variability across all temporal scales and seasons. ERA5, ERA5-Land, and CHELSA feature similar precipitation variability that is considerably lower than in CERRA and observations. In contrast, all datasets are more consistent with observations in terms of temperature variability. However, CERRA best represents temperature variability at daily and monthly scales, while ERA5, CERRA, and CHELSA are comparable in terms of the inter-annual temperature variability of most seasons as well as the annual maximum (minimum) of daily mean temperature. Only
ERA5-Land shows generally lower temperature variability than observations and all other datasets across all temporal scales.



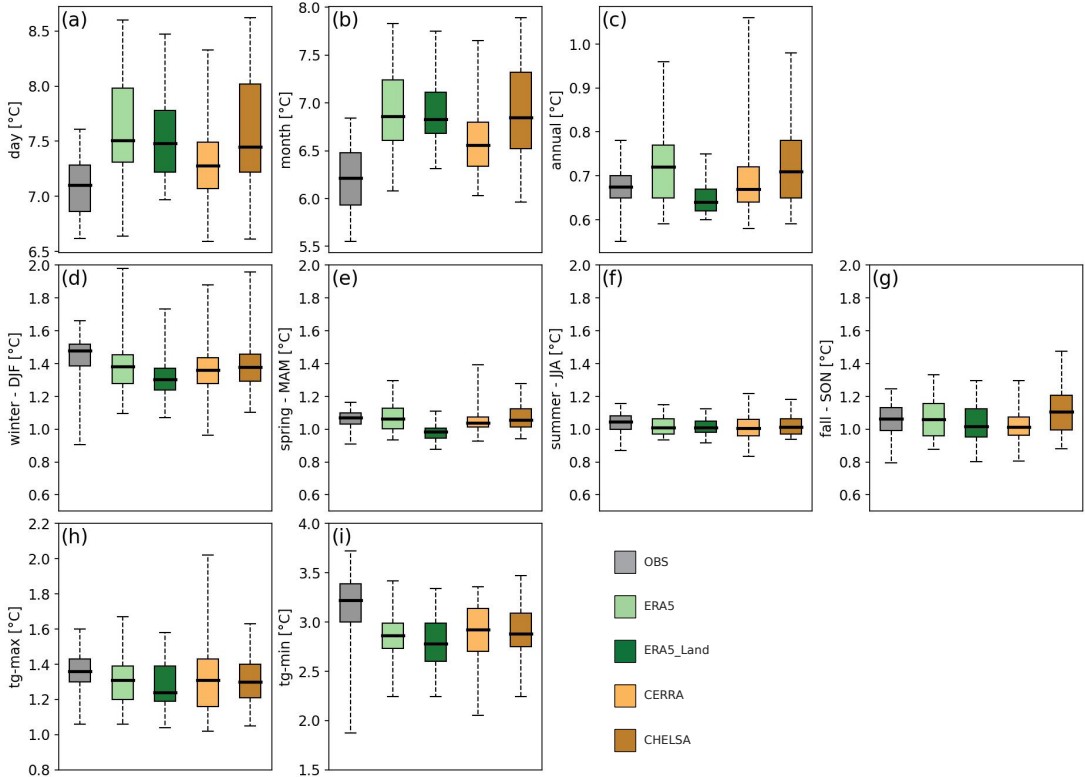

**Figure 6.** Comparison of daily to inter-annual temperature variability. Boxplots with (a) daily, (b) monthly, and (c) inter-annual temperature variability. (d)-(g) Show inter-annual variability for different seasons [(d) DJF, (e) MAM, (f) JJA, (g) SON]. (h) Shows the inter-annual variability of tg_max and (i) for tg_min. The boxplots represent variability across catchments and the colors indicate the datasets. Panels (a)-(c) show a different scaling on the y-axis due to the different aggregation periods. Temperature variability is expressed as the standard deviation of daily/monthly/annual values.

## 5.2 Trends

The presence of significant trends and trend magnitudes in precipitation and temperature metrics are best represented by CERRA, while the other datasets show inconsistent precipitation trends compared with observations. While CERRA shows the highest agreement with observed trends, it also tends to overemphasize the number of significant trends compared to observations and the other reanalysis datasets. Over the period 1986–2020, the observations show a general decrease in annual mean daily precipitation (Figure 7a), with approx. 20% of catchments showing significant (or weakly significant) trends (see Figure 7b, light grey bar) and 80% of catchments showing no significant trend (see Figure 7b, dark grey). ERA5, ERA5-Land and CHELSA show no significant trend in any catchment (Figure 7a,b). Therefore, they correctly identify all *no trend* catchments according to observations (see Figure 7b, medium light purple) and fail to represent trends for the 20% of the catchments that do show trends according to observations (see Figure 7b, hatched medium dark purple). CERRA shows a





general decrease in mean precipitation over time, with significant trends being detected in many catchments. About 50% of the catchments with a significant observed precipitation trend also show significant trends in CERRA (Figure 7b; ratio of the light grey and light purple bar). The remaining half of the catchments with significant observed trends remain undetected also by CERRA. CERRA shows considerably larger spread in trend magnitude across catchments than observations and the other

reanalysis datasets. As a result, it shows overall more catchments with significant trends where observations show no trend or both show opposing significant trends (see Figure 7, False: trends). These results are also consistent on the seasonal scale (see Figure S9).

The observed annual maximum 1-day precipitation sum (rx1d) shows considerably larger spread in trend magnitudes across all catchments compared to mean precipitation (Figure 7c). Depending on the catchment, we find positive or negative trends

in observed extreme precipitation. However, only 10% of the catchments show a significant (or weakly significant) trend in the observations (Figure 7d). ERA5, ERA5-Land and CHELSA again show no agreement with the significant trends in the observations (Figure 7d). They show a few individual catchments with *false trends* and largely agree on the catchments with *no trend*. Overall, they show a lower spread in trend magnitude across catchments than observations, with a median around zero (Figure 7c). CERRA agrees with the trends in observations for the majority of catchments, however, it again shows the largest

proportion of catchments with false trends. For the trends in mean annual solid precipitation, (Figure 7e,f) and the number of dry days (Figure 7g,h) this overall pattern remains: CERRA shows both the largest agreement in significant trends as well as the largest proportion of catchments with different trends than those in the observations, while ERA5, ERA5-Land and CHELSA show no overlap of catchments with significant trends in observations. For solid precipitation, all datasets –except for ERA5-Land which shows no change– represent the decreasing trends found in observations. For the number of dry days, neither

ERA5, ERA5-Land nor CHELSA show clear increases or decreases, which might be a reflection of the poor representation of dry days in these datasets (Figure 33g). In contrast, observations and CERRA tend to show an increase in the number of dry days.

All datasets agree on a significant increase in mean daily temperature (Figure 7i,j) and the temperature of the warmest day of the year (tg_max; Figure 7k,l) for all catchments, but not for the temperature of the coldest day (tg_min; Figure 7m,n). ERA5-

Land shows slightly weaker trends in mean temperature compared to observations, while ERA5, CERRA and CHELSA show stronger trends. The trend magnitude of temperature on the warmest day is comparable across all datasets. For the temperature of the coldest day, trend magnitudes in ERA5, CERRA and CHELSA are comparable with observations, while ERA5-Land underestimates the trend magnitudes. However, trends in temperature on the coldest day are largely non-significant: only 20% of catchments in the observations show a significant increase (Figure 7m,n). Among all datasets, CERRA agrees the most with

observed temperature trends as it shows both similar trend magnitudes and high agreement in true trends, as well as only a small number of false or undetected trends. The other datasets show a lower agreement in terms of true trends and a slightly larger proportion of false and undetected trends.

The number of cold days (Figure 7o,p) decreases in all reanalysis datasets and observations and this trend is significant in the majority of catchments. ERA5, CERRA and CHELSA show good agreement for the catchments with a significant trend

in observations and only a small proportion of undetected trends. However, all three datasets overestimate the number of



significant trends compared to observations (i.e., false trends). ERA5-Land shows the weakest decrease in the number of cold days as well as the lowest agreement with observations as it shows a larger number of catchments with undetected trends.

The general findings that CERRA has the highest agreement with observations with respect to the number of catchments with significant trends, false trends, as well as the best match in terms of trend magnitude, also applies for seasonal trends in
mean precipitation and temperature (Figure S9).





**Figure 7.** Comparison of trends in precipitation and temperature metrics. Violin plots show trend magnitudes (theil-sen slopes; 1986-2020) in observations (grey) and reanalysis datasets (colors) across all catchments for different precipitation (a, c, e, g) and temperature metrics (i, k, m, o). Stacked barplots show the correct spatial matching of significant trends or no trends by the reanalysis datasets (purple bars) compared to observations. Grey bars show the percentage of catchments with a significant trend (light grey) and no trend (dark grey) in the observations. Purple bars without hatching show the percentage of catchments in the reanalysis datasets with a correct matching of a significant trend (lightest coloring) or no trend (medium light). Hatched bars show the percentage of undetected (medium dark) or false trends (darkest coloring). Each bar represents the respective trends from the aligned violin plots. (a-b) mean daily precipitation (Pmean; mm), (c-d) maximum 1-day precipitation (rx1d; mm), (e-f) mean annual amount of solid precipitation (solidprcp; mm), (g-h) annual number of dry days (drydays; days), (i-j) mean daily temperature (Tmean; °C), (k-l) annual maximum of daily mean temperature (tg_max; °C), (m-n) annual minimum of daily mean temperature (tg_min; °C), and (o-p) number of cold days (colddays; days).





## 6 Representation of extreme events

### 6.1 Meteorological drought

The observed drought events in 2003 and 2018 show considerable differences in their spatial patterns, which all reanalysis datasets are able to capture. While ERA5, ERA5-Land and CHELSA overestimate drought severity and intensity, CERRA underestimates them. The drought in 2003 is characterized by many catchments showing dry (SPI6 ≤ -1) to very dry conditions (SPI6 ≤ -2) in the observations (Figure 8a) varying between -0.9 and -3.1 across catchments (e.g., Figure 8c, x-axis). Especially the central parts and southern parts of Switzerland show low SPI6 values and the largest cumulative precipitation deficits of more than 400mm (Figure 8a,b). In contrast, the drought 2018 affected fewer catchments (Figure 9a) and is characterized by a large spread of SPI6 values from 0.4 (no drought) to -2.8 (severe drought) across catchments (e.g., Figure 9c, x-axis). The center of the drought lay over the central and north-eastern parts of Switzerland, where cumulative precipitation deficits for several catchments were above 400mm in the observations (Figure 9b). The southern, eastern and northern parts of Switzerland were less affected by the 2018 drought. All reanalysis datasets capture these differences between the two drought events, that is the large spatial drought extent and small SPI6 spread between catchments in 2003 (Figure 8c-f), and the smaller drought extent and large SPI6 spread in 2018 (Figure 9c-f).

The reanalysis datasets ERA5, ERA5-Land and CHELSA show a high agreement with observations for the 2003 drought in terms of widespread drought conditions (SPI6 ≤ -1) (Figure 8c,d,f). However, they show a smaller catchment spread in SPI values than observations and have a tendency to overestimate drought severity (i.e., lower SPI6 values; points below the 1:1 line). Catchments which show an overestimation of SPI6 in the reanalysis datasets also show an overestimation of the precipitation deficit compared to the observed deficit by generally 20-80%, which can exceed 100% in individual catchments (Figure 8c,d,f, coloring of dots). CERRA generally underestimates drought severity and shows a larger spread of SPI6 values across catchment than observations. Furthermore, CERRA estimates SPI6 values above -1 for several catchments, indicating no drought, contrary to observations (Figure 8e). Although most catchment averages in CERRA agree with the observations on drought conditions and differences in the cumulative precipitation deficit are between +/- 20%, several catchments underestimate precipitation deficits by more than 40% compared to observations.



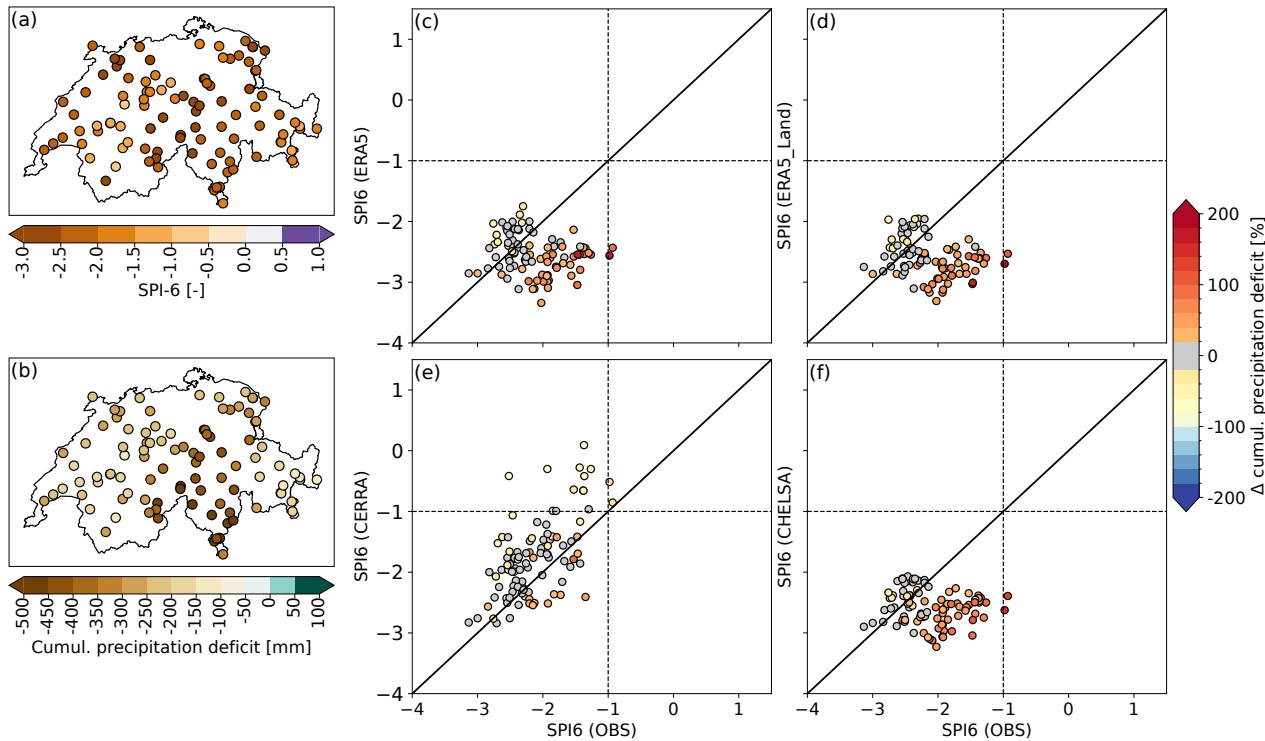

**Figure 8.** Comparison of the severity and intensity of the drought event 2003 for the different reanalysis datasets. (a) Shows the SPI6 catchment values (March-August) of the observations (i.e., severity). (b) Shows the cumulative precipitation deficit (mm) of the observations for the same period (i.e., intensity). The scatterplots in panels (c-f) show the four reanalysis datasets (y-axis) compared to the gridded observations (x-axis) for the period 1986–2020 and all catchments. The location of the dots shows the SPI6 values and the coloring of the dots shows the relative differences in the cumulative precipitation deficit (in percent compared to observed deficits). Orange and red coloring highlight larger deficits in the reanalysis, yellow coloring shows smaller deficits in the reanalysis, and blue colors show a surplus of precipitation rather than a deficit compared to observations. The dashed vertical and horizontal line indicate an SPI6 ≤ -1, which indicates moderate to severe drought conditions. The solid black line indicates the 1:1 line. (c) ERA5, (d) ERA5-Land, (e) CERRA, and (f) CHELSA.

The drought in 2018 is reasonably captured by ERA5, ERA5-Land and CHELSA, which show a similar spread of minimum and maximum SPI6 values as the observations, and a good match of catchments with low to high drought severity (Figure 9c,d,f). CERRA underestimates drought severity and shows a larger spread of SPI6 values (-3 to 1.3) than the observations (Figure 9e). A large number of catchments in CERRA shows undetected drought conditions (data points in the upper left quadrant; Figure 9e) or even a precipitation surplus (indicated by the blue coloring). Catchments which show no drought

condition in the observations also show no drought condition in CERRA, which shows some overall agreement despite some large deviations in event magnitude and several undetected drought occurrences.



Beyond the two drought years of 2003 and 2018, all reanalysis datasets agree well with the full distribution of observed SPI6 values (Figure S10), which indicates a correct temporal match of drought and no-drought conditions despite some apparent biases.

In summary, all four datasets capture differences between the two drought events, with the 2003 drought being better represented than the one in 2018. ERA5, ERA5-Land, and CHELSA overestimate drought severity and intensity in 2003 and partly under- and overestimate it in 2018, while CERRA overall underestimates the magnitude of both drought events.

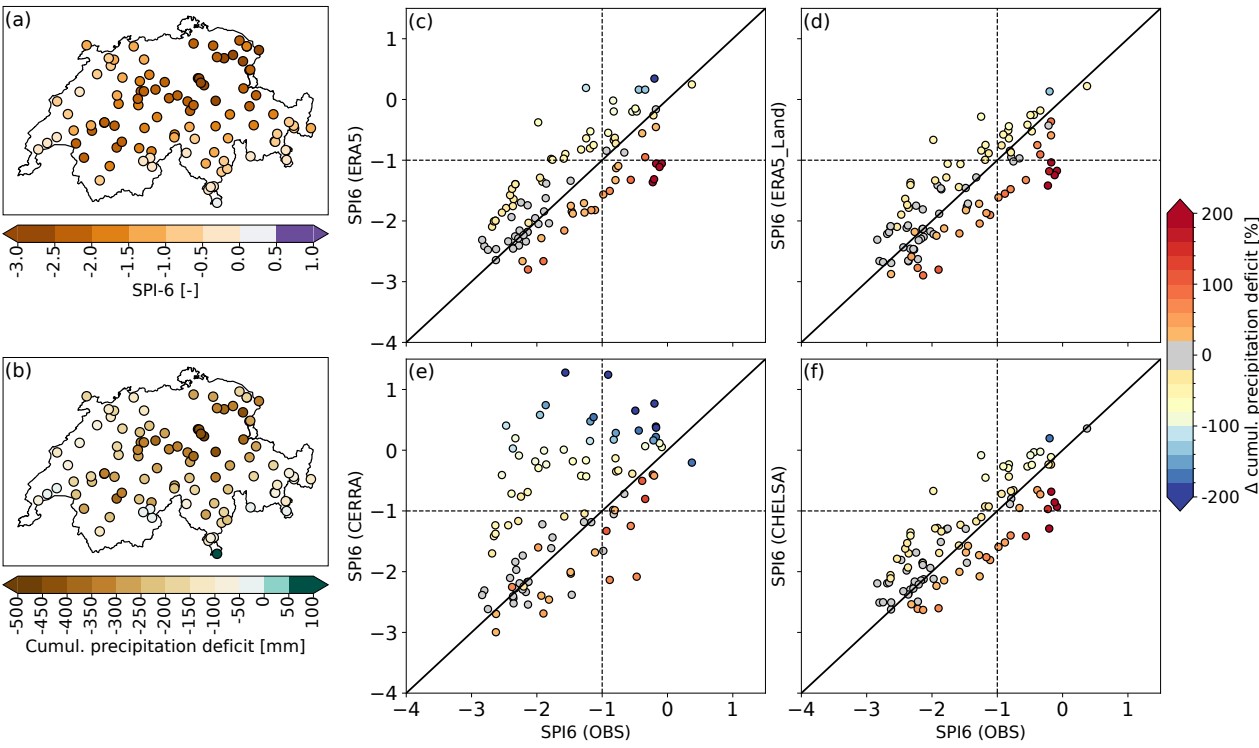

**Figure 9.** Same as figure 8 but for the drought 2018

## 6.2 Extreme precipitation

The intensity, severity and spatial structure of the three extreme precipitation events in 1999 and 2005 are captured by all
reanalysis datasets, with CERRA showing the highest agreement with observations, and the other datasets showing larger spatial extents than the observations and some overestimation and underestimation of event intensity locally. The 1999 extreme precipitation event was a sequence of two consecutive extreme precipitation events occurring only 10 days apart, with the major flood peaks observed during the second event. Both of these events were centered on the north-eastern part of Switzerland (Figure 10a-d). The first event on May 12th (end of accumulation period) covered a larger area with catchments showing standardized precipitation indices above 2 (Figure 10a) and precipitation intensities of 100-160 mm (Figure 10b). The second
event (May 22nd) had a smaller extent and was less intense (Figure 10c,d). All datasets are generally able to reproduce this event




sequence in 1999. The catchments showing high intensity rainfall in the observations are also affected by extreme precipitation in all four reanalysis datasets (see the large number of points in the upper right corner of Figure 11a-h). Thereby, the first precipitation event in 1999 (Figure 11a-d) is represented well by all datasets, with only a few catchments showing extreme

precipitation instead of moderate or low precipitation (upper left quadrant in Figure 11), or vice versa (lower right quadrant). CERRA generally matches the observations well but shows a clear underestimation of severity and intensity over the southern most catchments, which only received little to no precipitation. The second precipitation event in 1999 is partly overestimated by ERA5, ERA5-Land and CHELSA, which show a larger number of catchments with standardized precipitation above 1.5 than the observations (Figure 11e,f,h). In catchments with the highest event severity in observations, all three reanalyses (ERA5,

ERA5-Land and CHELSA) slightly underestimate event severity and intensity, while precipitation in catchments with low and moderate precipitation in the observations is largely overestimated. They show a larger spatial extent for the precipitation event than the observations, mainly in the south-east (towards the Engadin) and the south (Ticino). CERRA clearly shows a better match with observations than the other datasets (Figure 11g). It matches the geographical center of the event very well and only shows very small biases in the severity and intensity of the event.

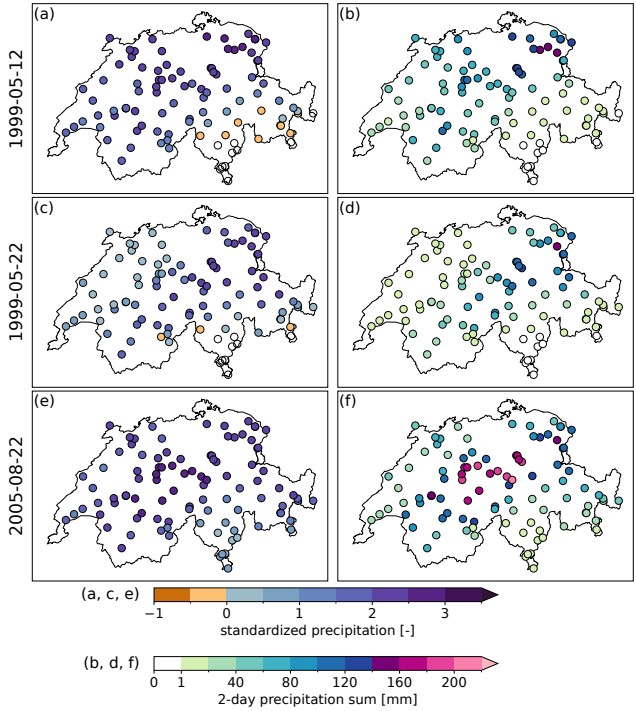

**Figure 10.** Maps of the severity and intensity of extreme precipitation events related to selected major floods in Switzerland based on gridded observations. The maps show the two extreme precipitation events in May 1999 (11–12 May 1999, a-b; 21–22 May 1999, c-d) and the single event in August 2005 (21–22 Aug 2005, e-f). Panels (a), (c) and (e) show the severity of the events expressed as the standardized 2-day precipitation sums. Panels (b), (d) and (f) show the intensity of the events expressed as the 2-day precipitation sum (mm).



The extreme precipitation event in 2005 was severe in most parts of Switzerland except the South, with intensities of 160 mm and more in large parts of central Switzerland, and slightly lower intensities in the West and East (Figure 10e,f). CERRA shows a very good agreement with observations in terms of severity and intensity (Figure 11k), and the geographical location of the center of the event. ERA5, ERA5-Land, and CHELSA slightly underestimate event severity but overestimate spatial extent compared to observations. All three datasets slightly underestimate event intensity and severity in those catchments which

show the highest severity in observations, and overestimate the severity and intensity in locations with low to moderately high precipitation (with approx. a standardized precipitation of 1.5).

In summary, CERRA shows the smallest biases of all four datasets for flood-triggering extreme precipitation, in particular for the most severe intensities. While ERA5, ERA5-Land, and CHELSA tend to overestimate the standardized precipitation

and precipitation accumulation for catchments with low to moderate precipitation, CERRA seems to slightly underestimate precipitation in these catchments. The spatial event extents are best represented by CERRA and and overestimated by ERA5, ERA5-Land, and CHELSA compared to observations.



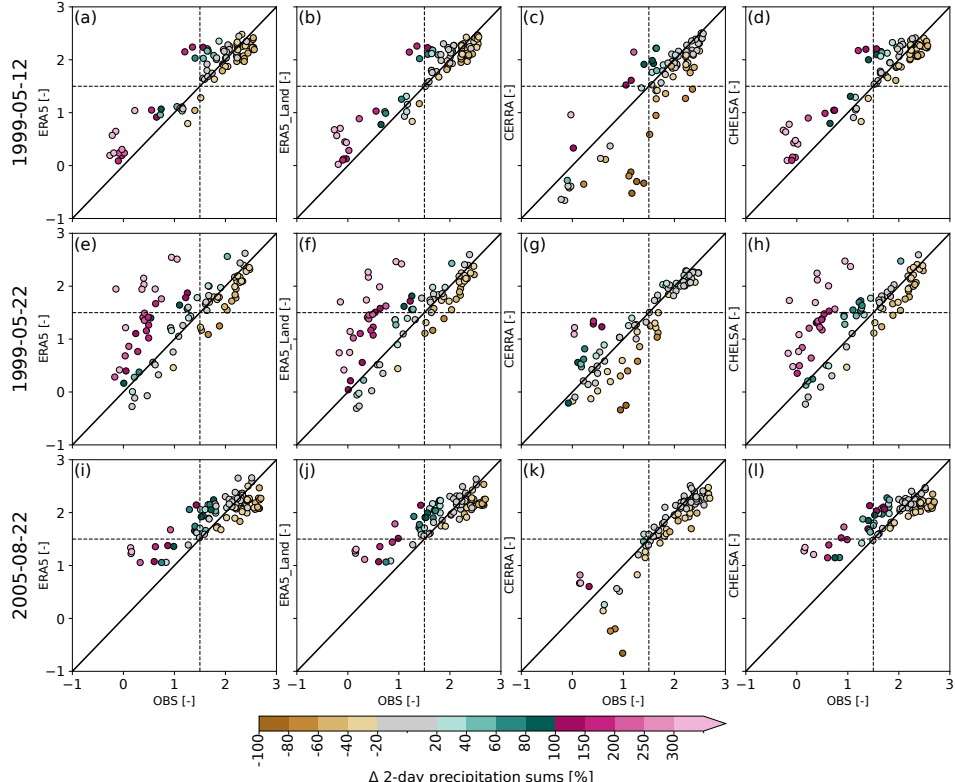

**Figure 11.** Comparison of the severity and intensity of extreme precipitation events related to selected major floods in Switzerland. Scatterplots compare the two extreme precipitation events in May 1999 (11–12 May 1999, a-d; 21–22 May 1999, e-f) and the single event in August 2005 (21–22 Aug 2005, i-l). Event severity is expressed as the standardized 2-day precipitation sums of the four reanalysis datasets (y-axis) and compared to the respective standardized precipitation of observations (x-axis, values from Figure 10(a, c, e)). Dot locations in the scatterplots show the standardized precipitation. For illustration purposes, the +1.5 standard-deviation is indicated by the dashed vertical/horizontal lines indicating heavy to extreme precipitation events and the 1:1 line for a perfect match. The coloring of dots in all panels shows the relative difference (%) in the 2-day precipitation sums compared to observations (values from Figure 10(b, d, f)). Brown colors indicate less precipitation than observed, grey precipitation in the range of +/-20%, and green and pink colors values above. Each dot represents one catchment.

# 7 Discussion

## 7.1 Climate metrics best represented in high-resolution datasets

Our results show that CERRA clearly improves the representation of precipitation metrics compared to the other reanalysis datasets (Figure 2, 3), likely thanks to the assimilation of data from additional precipitation stations within the MESCAN regional precipitation analysis. In contrast to CERRA, the other reanalysis products are not assimilating any precipitation observations; except for ERA5, which includes a regionally and temporally limited precipitation assimilation over the contiguous



U.S.A. through a radar-gauge product since 2010 (Hersbach et al., 2020). This highlights how closely model performance is linked with the availability of high quality data and dense station networks. Even though CERRA, among all reanalysis datasets, best represents most of the precipitation metrics, it shows some deficiencies in representing the studied drought events in 2003 and 2018 (Figure 8, 9). While the large-scale SPI pattern over Switzerland is simulated quite well, CERRA seems to simulate spurious local-scale precipitation events over regions with complex topography, which can locally alleviate the large-scale drought signals (Figure S11). This is an issue across the European domain, including Switzerland and the Apennine Mountains in Italy, where CERRA suffers from precipitation artefacts with bubble structures (Figure S12). We believe that these artefacts result from a lack of sufficient data for data assimilation over complex terrain, leading to a stronger reliance on the underlying numerical weather prediction model (HARMONIE-ALADIN). In the context of drought, a few short and heavy precipitation events can offset the presence of a drought locally, as seen in CERRA. Similar performance deficiencies of CERRA over regions with scarce station networks have also been highlighted by Le Moigne et al. (2021) over northern and eastern Europe.

Our finding that CERRA represents snowfall metrics well and ERA5-Land does not (Figure 4) is in agreement with Monteiro and Morin (2023), who in contrast to our results found a good performance for ERA5. This difference may be related to the scale difference between the two analyses. While our comparison focused on small to medium sized catchments, Monteiro and Morin (2023) looked at four large regions in the Alps, where higher resolution might be less crucial than at the catchment scale. Our results show strong differences between ERA5 and ERA5-Land in terms of temperature derived snowfall characteristics. While ERA5 clearly underestimates solid precipitation, ERA5-Land clearly overestimates it. These differences are likely driven by the considerable temperature differences between the two datasets as differences in precipitation are marginal. Generally, ERA5-Land shows lower temperatures and temperature variability than ERA5 and observations, especially in winter and spring. Further, ERA5-Land shows larger biases at lower and higher elevations and simulates more cold days than ERA5, which explains the overestimation of solid precipitation. Even though CHELSA shows similar biases as ERA5 for many of the climate metrics, the higher resolution of CHELSA seems to clearly improve the representation of snowfall compared to ERA5 (Figure 4). The snowfall overestimation by CHELSA at higher elevations might not be as severe as quantified, because the benchmark observations are known to underestimate solid precipitation at higher elevations ($> 1500$ m) by up to 30% (Bandhauer et al., 2021)).

All reanalysis datasets, except CERRA, clearly underestimate precipitation variability across all time scales (Figure 5). On daily timescales, this might be explained by the overestimation of low to moderate precipitation by the reanalysis datasets and the underestimation of heavy precipitation intensities, which leads to a smaller range of daily precipitation variance or a flattening of the intensity curve. However, for the underestimation of monthly and inter-annual variability, this explanation is likely not valid. The reasons for the underestimation of monthly to inter-annual variability could instead be model driven and caused by deficiencies in representing orographic effects or uncertainties in the data assimilation. Monteiro and Morin (2023) have shown no underestimation of precipitation variability by any of the datasets at the scale of the entire Alps. This suggests that the datasets agree well with observations at larger scales but not at local scales, where temporal variability is underestimated by most datasets.





Our results show that the studied extreme precipitation events have a larger spatial extent in ERA5 and ERA5-Land than in the other reanalyses datasets and the observations (Figure 11). This might result from their coarser grid resolution, which

has the effect that single grid cells influence the precipitation averages of multiple catchments. However, CHELSA, which has a much finer spatial resolution, shows the same behavior, likely because the CHELSA downscaling approach conserves the mass of precipitation fluxes at the original coarser grid resolution of the W5E5. This means that when we average CHELSA values over catchments, we move closer to the original coarser grid resolution. This highlights that the actual spatial scales of these datasets are likely coarser than their grid spacing, especially for the interpolated or statistically downscaled datasets

(ERA5-Land and CHELSA). This also applies to the gridded observations, which depending on the number of stations have an effective resolution of 15-20 km rather than 2 km (MeteoSchweiz, 2021a) or even coarser for intense convective precipitation (Frei and Isotta, 2019). Further, the results suggest that the spatial resolution of the ERA5 products is still not refined enough for heavy precipitation events, especially in complex terrain. Lavers et al. (2022) have found similar overestimation of the spatial extent of the heavy precipitation event during storm Alex in October 2020 on the northern side of the Alps, which they

attributed to a poor representation of the orography in ERA5 leading to a larger moisture influx from the south.

## 7.2 Spatial downscaling vs. dynamical downscaling

As all reanalysis datasets depend to varying degrees on the ERA5 reanalysis but use different downscaling techniques, we can assess whether statistical (i.e., CHELSA or ERA5-Land) or dynamical downscaling (i.e., CERRA family) leads to an improved representation of the hydro-meteorology compared to coarser resolution datasets (ERA5). Statistical downscaling did

not substantially improve the representation of temperature and precipitation metrics (Figures 2, 3). However, the statistical downscaling technique employed by CHELSA improved the representation of snowfall metrics (Figure 4), likely due to the refinement of resolution (1 km). This allows for the representation of locally varying elevation features, which play a key role in snow formation. ERA5-Land, which partly uses interpolated meteorological variables from ERA5, shows that a pure interpolation without any additional adjustments is not leading to any improvements in the representation of precipitation metrics,

compared to statistical downscaling with additional adjustments. In addition, the 9 km resolution of ERA5-Land might still be too coarse to refine the topography. In contrast to precipitation, ERA5 and ERA5-Land show considerable differences for temperature, which might be explained by the lapse-rate adjustment in ERA5-Land (Dutra et al., 2020), but might also be influenced by differences in soil-moisture interactions affecting evapotranspiration and energy fluxes which in turn influence temperature (Hersbach et al., 2020; Muñoz-Sabater et al., 2021; Scherrer et al., 2022). Our analyses do not allow for clear

conclusions on the isolated effect of dynamical downscaling because the analyzed precipitation data from CERRA in our study (i.e., CERRA-Land) relies on dynamical downscaling and an additional precipitation assimilation. However, CERRA(-Land) clearly shows that combining information from a reanalysis with observations benefits the overall representation of climate metrics even when the resolution remains the same. To disentangle the influence of the dynamical downscaling from the influence of the MESCAN regional precipitation system in CERRA(-Land), future work would need to include the precipitation

data from the CERRA high-resolution dataset, for which no additional precipitation information has been assimilated, and compare it to the precipitation data from CERRA with precipitation assimilation (i.e., CERRA-Land). The results from Ridal



et al. (2024) suggest that the dynamical downscaling, even without the precipitation assimilation, leads to a clear improvement in the skill of precipitation with a lower false alarm rate and lower RMSE values compared to ERA5. Further, they show that the precipitation from CERRA-Land, i.e. as used in our analysis, yields even greater skill compared to ERA5. This suggests that the overall improvement by CERRA-Land likely results from a combination of the added-value of the dynamical downscaling and the post-processing of the regional precipitation analysis MESCAN. Similarly, Bollmeyer et al. (2014) suggested that another dynamically downscaled regional reanalysis –the COSMO-REA6–, can show an improved representation of precipitation compared to its parent global reanalysis datasets (i.e., ERA-Interim) and that the added-value is especially pronounced in the case when the regional reanalysis included additional data assimilation.

## 7.3 Limitations

We acknowledge that the 35-year time period studied (1986-2020) may be too short to robustly estimate the real spread of inter-annual variability and to robustly detect trends. Especially for precipitation but also for temperature, a large number of samples is needed to adequately gauge inter-annual variability (Wood et al., 2021; Maher et al., 2020). However, as the reanalysis datasets are constraint by observations, the reanalyses and observations should both represent the same large-scale variability and this constraint can partly be neglected. Nevertheless, the presented estimates of variability, based on 35 years of data, might deviate from the variability as estimated by longer periods. Besides variability, the time period length also influences the detection of trends as small signal to noise ratios can mask trends. Further, any trend analysis is time period-sensitive and are only a snapshot in time. For example, our trend analysis starting in 1986 yields decreasing trends in observed extreme precipitation (Figure 7c), however, starting in 1901 yields clearly increasing trends in observations (see e.g., Scherrer et al., 2016). In addition, the time varying quantity and quality of stations, which are in varying degrees used to generate the studied datasets and observations might lead to spurious or artificial trends Monteiro and Morin (2023). This could be a reason for the large number of significant or opposite trends in the CERRA dataset.

One main drawback of the current version of CERRA is the short simulation period (1984–2021). However, there are plans for an extension of the dataset to the present and a back-extension to 1961 (Ridal et al., 2024). Once the long time period is available, the drawbacks coming from the short simulation period can partly be neglected. However, the good performance of CERRA over the recent period, which can in part be attributed to the large number of station data included in the MESCAN regional precipitation analysis system, needs to be re-assessed and confirmed for the back extension, where abundant and high-quality station records are rarer.

The choice of the gridded observational product from MeteoSwiss might have influenced the results of our evaluation. Kotlarski et al. (2017) have shown that such influence is rather weak for temperature, but larger for precipitation. The precipitation observations (RhiresD) used in this study tend to overestimate light precipitation and underestimate strong precipitation (MeteoSchweiz, 2021a; Kotlarski et al., 2017). This means that the underestimation of extreme precipitation metrics in ERA5, ERA5-Land, and CHELSA is likely even larger than shown here. In general, gridded observations suffer from uncertainties not only from measurement errors (Kochendorfer et al., 2017) but also from the interpolation of station data (Frei and Isotta, 2019). The interpolation uncertainty is thereby influenced by varying station densities or the lack of representative stations,



especially at high altitudes (Frei, 2013). For the gridded temperature dataset TabsD, Frei (2013) show seasonally varying interpolation errors with larger biases in winter, which can reach mean absolute errors of 3°C and more, and smaller errors in summer. This means that the dataset differences, especially in winter, could in reality be larger or smaller depending on the location. To account for these uncertainties, more observational datasets would be required, although most gridded datasets

would rely on more or less the same station network, which would then only allow for disentangling the uncertainties from the interpolation scheme. As the interpolated fields are only one possible realization of the real spatial distribution, one could use ensembles of the same gridded product to estimate the interpolation uncertainty (e.g., Frei and Isotta, 2019). However, not many ensemble products for gridded observations exist and for example Bandhauer et al. (2021) showed that such an ensemble might not represent the full spread of interpolation uncertainty.

## 8 Conclusions and recommendations


In this paper, we conducted a comprehensive spatio-temporal evaluation of four state-of-the-art reanalysis datasets (ERA5, ERA5-Land, CERRA(-Land), and CHELSA-v2.1) for different precipitation, temperature and snowfall metrics over complex terrain by comparing them to gridded observations. We analyzed differences in mean and extreme climatologies, daily to inter-annual variability, as well as the consistency in long term trends. Further, we compared the representation of extreme events,

namely the intensity and severity of the 2005 and 2018 drought, and the 1999 and 2005 floods in Switzerland.

While all reanalysis products capture mean and extreme temperature characteristics similarly well, the CERRA dataset captures mean and extreme precipitation characteristics best likely to its data assimilation scheme (Figure 12), making it a good choice for hydrological impact studies. CERRA also provides the best representation of precipitation variability at all timescales and temperature variability at daily and monthly scales. In contrast, the other reanalysis datasets generally consider-

ably underestimate precipitation variability, while representing temperature variability well – an exception being ERA5-Land, which also underestimates temperature variability. Overall, all four reanalysis datasets are able to represent extreme dry and wet meteorological conditions of varying severity and intensity as illustrated for the droughts in 2003 and 2018 and the extreme precipitation events in 1999 and 2005. CERRA best represents wet extremes, while the other datasets overestimate spatial extents, severity and intensity in catchments with low to moderate precipitation characteristics. However, CERRA overall un-

derestimates the severity and intensity of the drought events studied, which seems to be the major limitation of this dataset. For applications where the representation of snow matters, we can clearly recommend the use of the high-resolution products CERRA (5.5 km) and CHELSA (1 km). Both datasets show the best overall agreement with observations for the fraction of solid to total precipitation and the total amount of solid precipitation. Both high- and low-resolution reanalysis datasets struggle with representing trends in temperature and precipitation characteristics. Again CERRA stands out as the only reanalysis

that can capture some of the observed temperature and precipitation trends, while all other models show hardly any significant trends. In conclusion, CERRA seems to be the best reanalysis choice for hydrological impact studies that rely on precipitation, temperature, and their interplay as it captures their mean characteristics, variability and extremes well, with the exception of distinct droughts. However, the other reanalysis datasets also show a satisfactory or good performance for some of these



properties and are a viable option for the analysis of hydro-meteorological extremes, as long as their limitations and biases are

considered.

**Figure 12.** Summary of results and dataset limitations

*Code and data availability.*  The reanalysis datasets ERA5, ERA5-Land, and CERRA are all freely available through the Copernicus Climate Change Service (C3S) Climate Data Store (CDS). ERA5: 10.24381/cds.adbb2d47 (Hersbach et al., 2023); ERA5-Land: 10.24381/cds.e2161bac (Muñoz Sabater, 2019); CERRA high resolution: 10.24381/cds.622a565a (Schimanke et al., 2021); CERRA-Land: 10.24381/cds.a7f3cd0b (Verrelle et al., 2022). The CHELSA dataset is available through EnviDat: https://www.doi.org/10.16904/envidat.228 (Karger et al., 2021a).

CAMELS-CH is available through Zenodo: 10.5281/zenodo.7957061 (Höge et al., 2023a). The *xclim python package* v0.44.0 for the climate indicators is available through Zenodo: https://doi.org/10.5281/ZENODO.8075481 (Bourgault et al., 2023a).



*Author contributions.* RRW conceptualized and performed the majority of the analysis, and wrote the first draft of the manuscript. JJ contributed to the drought analysis, supported the data pre-processing, and contributed to the writing of the manuscript. AvH supported to the data pre-processing and contributed to editing and reviewing of the manuscript. JG contributed to the flood event analysis and to editing and reviewing of the manuscript. DS contributed to editing and reviewing of the manuscript. MIB supported the data analysis and contributed to the writing, reviewing and editing of the manuscript and acquired the funding for this project.

*Competing interests.* One of the co-authors is a member of the editorial board of Hydrology and Earth System Sciences.

*Acknowledgements.* We thank Ruth Lorenz for her vital support in data curation and data management. We acknowledge funding from the Swiss Federal Office for the Environemnt (FOEN) through the HydroSMILE-CH project and the Swiss National Science Foundation (SNSF) through project 'Predicting floods and droughts under global change' (PZ00P2_201818).



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
