# Peer review of "Comparison of high-resolution climate reanalysis datasets for hydro-climatic impact studies"

_EGUsphere, 2024_

## Referee Comment (RC1)

**Review of the manuscript "Comparison of high-resolution climate reanalysis datasets for hydro-climatic impact studies"**

In this study, Wood et al. compared how four different reanalysis datasets (ERA5, ERA5-land, CERRA, and CHELSA) reproduce the observed average, extremes, and trends in precipitation and temperatures over a set of catchments located in Switzerland. These datasets differ in their spatial and temporal resolution, as well as in their calculation methods. The results show that CERRA generally outperform the other datasets, likely due to the data assimilation, which make it the most reliable dataset for hydrological impact studies.

Overall, the manuscript is well written and structured, the objectives and methods are clearly defined. It addresses relevant questions on the quality of the dataset that we use in hydrology studies (among other disciplines). Therefore, I think that the manuscript is a useful contribution to the HESS journal. Nevertheless, the comparison of some of the metrics and the consistency of the figures could be further improved before publication (see the comments below).

**Specific comments**

**Abstract**

Line 10: I suggest to rephrase the sentence about drought and floods since the datasets do not properly simulate flood and drought events, but rather the conditions (precipitation and temperature) that lead to droughts and floods.

**Introduction**

Lines 28-29: Could you provide a reference for the statement "Among these, the highest quality datasets are those relying on spatially interpolated observations."?

Lines 65-66: "many applications require even higher resolution data". Could you mention a few examples and explain why higher resolutions is needed?

Line 76: I would suggest specifying the type of drought here (I guess hydrological), since one could also be interested in understanding the drivers of atmospheric drought.

Lines 83-91: This paragraph is difficult to follow, as we move from one study to another with different objectives and spatial scales. Also, the conclusions of these studies are missing. We understand that they focused on temporal variations in precipitations (including snow) and their impacts streamflow events, but it could be more explicit.

**Data and Methods section**

Overall comment: the criteria for selecting the four datasets are missing. Why did you select these four datasets? As you mentioned previously, Dura et al (2024) compared seven datasets, why not including all of them in your study?

Line 182: I wonder why you did not used precipitation and temperature provided by the CAMELS-CH? Do they come from the same products?

Line 186: In my opinion, the first analyses focus more on comparing the spread of the climate metrics (figures 2 and 3), than on the absolute or relative differences between the metrics, which are presented in the supplementary material (more about that in the comments of the results section).

Lines 192-196: I didn't find this comparison between time series-based and metric-based approaches in the results. Did you observe differences between these approaches? Anyway, I would suggest to stick to the metric-based approach in the manuscript.

Line 198 and 204: Are the annual mean and other metrics such as wetdays computed over hydrological years?

Line 203: why keep both wet and dry days metrics? Aren't they 100% correlated? The wetdays is shown in Figure 2 (while drydays isn't), and reversely drydays is shown in Figure 7 (but wetdays isn't). I suggest to stick to one or the other of these metrics.

Lines 211-212: How different are the estimations of snowfall (re)computed in your study from those provided by the datasets?

Table 2: The metrics for the mean air temperature is missing in the table. Also, since you used the colddays, why not including hotdays as well (perhaps with a threshold adapted to your catchment sample)?

**Results section**

Overall comment: In sections 4.1, 4.2, 5.1, the analyses and the figures compare the distribution of the metrics (the boxplots), including all the catchments. From these results we can not conclude that the reanalysis products reproduce well (or not) the different metrics for each catchment. Therefore, I would suggest to plot the distribution of the bias (between obs. and reanalysis data computed for each catchment) instead. This would also help to estimate which dataset over- or under-estimate the metrics (which is difficult to assess on the figures in the manuscript). Or, even better in my opinion, to show the scatter plot of the metrics 'simulated' vs 'observed', as you did for the SPI (e.g. Figure S10). From these, we could assess the correlation between the reanalysis and the observed metrics, as well as their respective spread.

In addition, I don't understand the different types of representations of the results between:

   a) the box plots in Figures 2, 3 and 5,
   b) the scatter plots in Figures 4, 8 and 9,
   c) and the box + violin plots in Figure 7.

a) and b) aimed at the same objective, which is to compare the datasets, why are the results presented in different ways?

c) shows the distribution of the trends, but why adding violin plot? It is redundant with the boxplot (and previous figures did not display violin plots).

Lines 278-280: "The positive reanalysis biases are particularly evident in catchments at high- and low-elevations (>2000 and < 1000 m.a.s.l.)" It seems that you have more catchments in these high and low altitudes, perhaps you could provide a histogram of the catchment altitudes?

Line 305: The differences in metrics are described in function of the altitude of the catchments, perhaps you could define the thresholds for low/mid/high altitude catchments once, and refer to these groups of catchments when describing the results?

Lines 312-315 How is this short section about the relationship between the fraction of snow and altitude relevant for the study (Figure 4e)?

Line 319: The 100% of under- and over-estimation could be related to catchment with very low fraction of snow.

Line 332: Did you mean that CHELSA overestimates _solid_ precipitation? (the world "solid" is missing)

Figure 4: The figure would be easier to read with the name of the products on the plot (ERA5, ERA5Land …).

Line 337: I suggest to delete the parentheses about "variability from one day to the other" since this is not what the coefficient of variation (or the standard deviation) reflects. You could rank the daily precipitation from the highest to lowest precipitations during a year (which would result in low day to day variation) and get the same standard deviation value.

Figure 5: Rx1day and Rx5day are shown in the figure 5 but not discussed in the text. Moreover, the figure compares the distribution of the metrics, therefore I suggest to change the legend of the figure accordingly (perhaps to something like "Comparison of the distribution of daily to inter-annual precipitation variability"), the same apply for previous figures 2, 3, 5, 6, 7.

Line 349: I suggest to add a reference to the figure 6 and the end of this sentence.

Line 385: On figure 7d, the proportion of catchments with significant trend in Rx1day seems closer to 20% than 10%.

Line 394-396: The reasons (poor representation of dry days) for the poor trends detection in those metrics should be moved in the discussion section.

Figure 7: This figure is hard to read. Why adding violin plot on top of boxplot? The labels of plots i-o are too close to plots b-h, and can easily be confused with the x-axis of the other plot. The legend for the significance of the trends feel too complicated. The information of interest are: is the trend significant? Is it increasing or decreasing? And do we observe it in the simulated data? Wouldn't it be easier to represent using: color (e.g. from blue to red) for sig. increasing/ no-trend / sig. decreasing in observed data, and then hatch the proportion of catchment in which the simulated data failed to reproduce the observed trend?

Lines 418-419: I suggest to add a reference to the figure 8 at the end of the sentence.

Figures 8 and 9: I find the figure S10 more interesting and more straight to the point than the figures 8 and 9 in the manuscript. I would suggest replacing Figures 8 and 9 by Figure S10.

Line 469: What are the results discussed here? Figures S11/S12?

**Discussion section**

Overall comment: One of the objectives of the study was to investigate how the datasets perform in complex terrain, with a focus on mountain region. This should be discussed more here, since you

have low/high altitude catchments, the comparison of the behavior of the dataset in different categories of catchment could be investigated in more details. Can we use these datasets for catchments in high altitudes? Why? Why not? What are the recommendations? How could we improve the datasets for these regions?

Line 490: I think that the statement that CERRA clearly improves the representation of precipitations metrics is too strong. The analyses show that it improves how the spread is reproduced over the sample of catchment, however there is no evidence that bias is lower at the catchment scale. What about the spatial correlations of the metrics (obs. vs. reanalysis datasets)?

Lines 494-495: I suggest to move the sentence about the precipitation assimilation of ERA5 in the method section (or to remove it) as it adds nothing to the discussion here.

Lines 499-500: Is there a reference supporting the effect of bubble structures in Europe?

Line 505: The snowfalls evaluated are not those that were originally provided in the datasets, I think that it should be reminded here that it was re-calculated for this study (Lines 212-213).

Lines 525-527: If datasets and observation agree well at larger scales, should we use them (instead of more refined datasets) when studying large catchments? Should we select different dataset based on catchment size?

Lines 542-544: As stated below (lines 555-556), the isolated effect of statistical and dynamical downscaling can not be assessed with these datasets, since CERRA uses data assimilation. This sentence should be reworded.

Lines 577-580: Bruno et al. (2023) have also clearly shown how trends (in catchment evapotranspiration) can vary depending on time periods. (Figure 1 in Bruno, G., & Duethmann, D. (2024). Increases in water balance-derived catchment evapotranspiration in Germany during 1970s–2000s turning into decreases over the last two decades, despite uncertainties. *Geophysical Research Letters*, 51, e2023GL107753. https://doi.org/10.1029/2023GL107753).

Line 597: Errors of > 3 °C during winter could also explain why the snowfall metrics are not well reproduced by some datasets.

**Conclusion**

What the word "variability" (Lines 615, 616) refers to is not totally clear to me: is it temporal variability? spatial variability? Both?

I think that the summary of the results could be shortened, in the favor of a focus on the strength and weaknesses of the datasets in mountainous areas (which is a gap of knowledge identified Lines 96-97). More recommendations on which dataset to use, where, and why (or why not) would be appreciated.

The last sentence is vague and could be removed since all datasets have limitations (CERRA included).

Figure 12: Could this figure be introduced in the discussion section? It is very useful, although the attribution of performance grade (poorly/satisfactory/well) seems very subjective. Perhaps you could explain how these performance grades were determined in the section on methods?

**Technical corrections**

Line 65: the name of the product and its spatial resolution are in different orders in this sentence: CERRA at 5.5 km, vs. 6 km COSMO-REA6. I would suggest to rephrase for consistency.

Line 104: "…the representation of  all of these components…"

Line 106: "To shed light on the question **of** which reanalysis products are most suitable…"

Line 133 and table 1: "…the Integrated Forecasting System Cy41r2 and covers the period from 1940 until the present". In the text, ERA-5 started in 1940, but in table 1 it started in 1950.

Line 192: "To calculate catchment averages **metrics**, we use two complementary…".

Line 208, Table 2, Figure 3: The metric for annual min and max of temperature are written as "tg_min" and "tg_max" in the text and figure 3, vs. "tg-min" and "tg-max" in table2.

Figure 3: Labels of the metrics: rx1day in the figure vs. Rx1day in table 2 (same thing for r99ptot).

Line 396: "Figure33g" -> Figure 3g

Line 486: "…by CERRA and  overestimated…"

Line 518: "(Bandhauer et al., 2021)". (there is an extra closing parenthesis)

Line 581: "or artificial trends Monteiro and Morin (2023)." I guess that the format of the citation is wrong here.

Line 634: "The CHELSA  **dataset** is available…"

---

## Author Comment (AC1)

**Response to Reviewer 1**

Dear Laurent,
We thank you very much for your positive and very constructive review. We appreciate the time you took to review our manuscript in such detail. We agree with most of your comments and decided to follow your recommendation to change some of the figures and give more detail to difference between elevation bins (especially figure 2, 3, 5 and 6). In that regard we switched from presenting the distribution of the metrics to focusing on the biases on the catchment level.

In the following your comments are in black and our response in green.

Overall, the manuscript is well written and structured, the objectives and methods are clearly defined. It addresses relevant questions on the quality of the dataset that we use in hydrology studies (among other disciplines). Therefore, I think that the manuscript is a useful contribution to the HESS journal. Nevertheless, the comparison of some of the metrics and the consistency of the figures could be further improved before publication (see the comments below).

Thank you very much for your positive assessment!

**Abstract**
Line 10: I suggest to rephrase the sentence about drought and floods since the datasets do not properly simulate flood and drought events, but rather the conditions (precipitation and temperature) that lead to droughts and floods.
Reviewer 2 had the same comment. We will rephrase this to "meteorological drought" and "extreme precipitation events".

**Introduction**
Lines 28-29: Could you provide a reference for the statement "Among these, the highest quality datasets are those relying on spatially interpolated observations."?
Lines 65-66: "many applications require even higher resolution data". Could you mention a few examples and explain why higher resolutions is needed?
We will provide additional references in the locations that you suggest.

Line 76: I would suggest specifying the type of drought here (I guess hydrological), since one could also be interested in understanding the drivers of atmospheric drought.
Yes, this should say "hydrological drought". We will clarify this.

Lines 83-91: This paragraph is difficult to follow, as we move from one study to another with different objectives and spatial scales. Also, the conclusions of these studies are missing. We understand that they focused on temporal variations in precipitations (including snow) and their impacts streamflow events, but it could be more explicit.
We tried to be concise here, but we might have been too minimalistic with the information. We will provide more details on the studies.

**Data and Methods section**
Overall comment: the criteria for selecting the four datasets are missing. Why did you select these four datasets? As you mentioned previously, Dura et al (2024) compared seven datasets, why not including all of them in your study?
Response to the dataset selection: Our interest are the newest generation reanalysis products at higher resolution. Hence, our choice of the ERA5 suite and its derivates. Compared to the older generation reanalysis (ERA-interim), ERA5 has not only been improved in spatial resolution but also

in physical representation. The focus on the higher resolution reanalysis meant that we a priori exclude the other widely used global reanalysis (i.e., MERRA2 and JRA-55) which only provide data at very coarse resolution (>50 km). Despite the short simulation period, the datasets CERRA and Chelsa were chosen as they are expected to be extended back in time, soon offering a valuable tool for long term analysis. Further, there is currently only a limited number of studies available that evaluated CERRA or Chelsa, which in our opinion meant that these datasets required a thorough comparison. We will add a sentence on the reasoning of data choice.

Response to the datasets in Dura et al (2024): The focus of Dura et al (2024) was on France which meant that their data selection was tailored to the availability over France. Three of the datasets in Dura et al (2024) are only available over France and one dataset is a CPM run that dynamically downscaled another RCM which in turn has been driven by ERA-interim. Therefore, these datasets either didn't provide any data over the catchments or it is a second order derivative of an old reanalysis product.

Line 182: I wonder why you did not used precipitation and temperature provided by the CAMELS-CH? Do they come from the same products?
The meteorological data in CAMELS-CH are based on the same data from MeteoSwiss as we used in our study. Other available meteorological data within the CAMELS-CH dataset are post-processed MeteoSwiss data from the PREVAH hydrological model. We chose to use the original MeteoSwiss products instead to have full flexibility on the analysis. Especially for the climate indicators, which we calculated on the grid cell first and then averaged over the catchments, required us to use the gridded data and not the timeseries data from CAMELS-CH.

Line 186: In my opinion, the first analyses focus more on comparing the spread of the climate metrics (figures 2 and 3), than on the absolute or relative differences between the metrics, which are presented in the supplementary material (more about that in the comments of the results section).
We agree with you. We took your advice and adapted the figures to show differences rather than the spread of the climate metrics. Please see the response to the other comments.

Lines 192-196: I didn't find this comparison between time series-based and metric-based approaches in the results. Did you observe differences between these approaches? Anyway, I would suggest to stick to the metric-based approach in the manuscript.
We didn´t plan to show a comparison between the two approaches. We decided to go with the two approaches as we think that the different analyses require different approaches. For the metrics mean precipitation and mean temperature, annual or seasonal, the difference between the approaches is only marginal. For the other metrics the different approaches lead to different results. We will extent the justification of the two approaches in the text.

Line 198 and 204: Are the annual mean and other metrics such as wetdays computed over hydrological years?
Only the snow related analysis is based on the hydrological year, other metrics are based on the calendar year. For the snow metrics we explicitly mentioned this in the text, but for the other metrics we didn´t. We will clarify this.

Line 203: why keep both wet and dry days metrics? Aren't they 100% correlated? The wetdays is shown in Figure 2 (while drydays isn't), and reversely drydays is shown in Figure 7 (but wetdays isn't). I suggest to stick to one or the other of these metrics.
As reviewer 2 also mentioned this, we will streamline the presentation of metrics and will select consistent metrics across figures. This means that we will likely remove dry days as a metric.

Lines 211-212: How different are the estimations of snowfall (re)computed in your study from those provided by the datasets?

We did not quantify this. We chose to use a consistent approach to separate liquid and solid precipitation as not all datasets provide snow height or snow water equivalent or the separation of liquid and solid precipitation. Further, in many applications (e.g., hydrological models or land surface models) snow is calculated within the model itself often following a similar approach using a temperature index approach.

Table 2: The metrics for the mean air temperature is missing in the table. Also, since you used the colddays, why not including hotdays as well (perhaps with a threshold adapted to your catchment sample)?

We will adapt the table after we streamline the set of metrics that we want to show in the manuscript. This means we will include the missing mean air temperature metric and might remove some other metrics. We chose to show cold days, as this is the threshold for the separation of liquid and solid precipitation, and therefore is a universal threshold. We could have shown hot days as well, but as you say we would have had to specify a catchment specific threshold. As we cover high elevated and low elevated catchments a "hot day" will require very different thresholds, which to some extent would be an arbitrary choice. Further, as we already compare many different metrics, we choose to not add more metrics.

**Results section**

As your next comment covers several topics, we will split your comment and reply to the individual topics.

Overall comment: In sections 4.1, 4.2, 5.1, the analyses and the figures compare the distribution of the metrics (the boxplots), including all the catchments. From these results we can not conclude that the reanalysis products reproduce well (or not) the different metrics for each catchment. Therefore, I would suggest to plot the distribution of the bias (between obs. and reanalysis data computed for each catchment) instead. This would also help to estimate which dataset over- or under-estimate the metrics (which is difficult to assess on the figures in the manuscript).

We fully agree with you. We will follow your recommendation and will adapt the figures accordingly to show biases instead of the distribution of the metric.

Or, even better in my opinion, to show the scatter plot of the metrics 'simulated' vs 'observed', as you did for the SPI (e.g. Figure S10). From these, we could assess the correlation between the reanalysis and the observed metrics, as well as their respective spread.

This would be informative, however by displaying the results in the style of Figure S10 would increase the number of figures dramatically. Each dataset would require its own panel (i.e., as in Figure S10), which would mean that if we want to retain all seasons in figure 2, we will require 40 panels, and 24 panels for figure 3. Therefore, we will keep the boxplots, but agree with showing the distribution of biases instead of climatological means.

In addition, I don't understand the different types of representations of the results between:
a) the box plots in Figures 2, 3 and 5,

To show multiple metrics in a concise way.

b) the scatter plots in Figures 4, 8 and 9,

Here, we compare individual metrics, hence we can allow for a more detailed presentation of the results.

c) and the box + violin plots in Figure 7.

Needed a way to compare (a) the magnitude in trends and (b) the consistency in trends. Both of this information need different ways of conveying the information.

a) and b) aimed at the same objective, which is to compare the datasets, why are the results

presented in different ways?

See our reply above. Figures 2, 3, and 5 include multiple metrics and therefore need a concise way of presentation. Figures 4, 8, and 9 show individual metrics per figure.

c) shows the distribution of the trends, but why adding violin plot? It is redundant with the boxplot (and previous figures did not display violin plots).

We will likely change the presentation of results in figure 7. We will certainly streamline the presentation and will change the violin plots to boxplots. See also the response to your specific comment on figure 7.

Lines 278-280: "The positive reanalysis biases are particularly evident in catchments at high- and low elevations (>2000 and < 1000 m.a.s.l.)" It seems that you have more catchments in these high and low altitudes, perhaps you could provide a histogram of the catchment altitudes?

Line 305: The differences in metrics are described in function of the altitude of the catchments, perhaps you could define the thresholds for low/mid/high altitude catchments once, and refer to these groups of catchments when describing the results?

The three elevation bins (high: >2000, mid: 1000-2000, and low: <1000 m) are almost equal in the number of catchments. We have adapted the figures to now also include the information on the three elevation bins. The boxplots are now overlaid with the median bias of catchments within each elevation bins (three different markers).

Lines 312-315 How is this short section about the relationship between the fraction of snow and altitude relevant for the study (Figure 4e)?

To give some context.

Line 319: The 100% of under- and over-estimation could be related to catchment with very low fraction of snow.

Yes, certainly.

Line 332: Did you mean that CHELSA overestimates solid precipitation? (the world "solid" is missing)

Yes, thank you for noticing. We added "solid" for clarification.

Figure 4: The figure would be easier to read with the name of the products on the plot (ERA5, ERA5Land ...).

Agree, we will adapt the figure accordingly.

Line 337: I suggest to delete the parentheses about "variability from one day to the other" since this is not what the coefficient of variation (or the standard deviation) reflects. You could rank the daily precipitation from the highest to lowest precipitations during a year (which would result in low day to day variation) and get the same standard deviation value.

We will remove the misleading information.

Figure 5: Rx1day and Rx5day are shown in the figure 5 but not discussed in the text. Moreover, the figure compares the distribution of the metrics, therefore I suggest to change the legend of the figure accordingly (perhaps to something like "Comparison of the distribution of daily to inter-annual precipitation variability"), the same apply for previous figures 2, 3, 5, 6, 7.

We will add the information on the variability of Rx1d and Rx5d in the text.

Further, following your comments and recommendation we changed figures 2, 3, 5 and 6 to show biases instead of the distribution of the climate metrics. However, in figure 7, we would like to keep the comparison of the distribution of trends rather than their biases, because due to the short time period the trend estimation itself might not be as reliable, as we also mention in the discussion, and therefore we would like to relax the comparison a bit and not show biases.

Line 349: I suggest to add a reference to the figure 6 and the end of this sentence.
Added the reference.

Line 385: On figure 7d, the proportion of catchments with significant trend in Rx1day seems closer to 20% than 10%.
True, we adapted this accordingly.

Line 394-396: The reasons (poor representation of dry days) for the poor trends detection in those metrics should be moved in the discussion section.
Yes, this could fit into the discussion section. We will see whether this fits better in the discussion or not.

Figure 7: This figure is hard to read. Why adding violin plot on top of boxplot? The labels of plots i-o are too close to plots b-h, and can easily be confused with the x-axis of the other plot. The legend for the significance of the trends feel too complicated. The information of interest are: is the trend significant? Is it increasing or decreasing? And do we observe it in the simulated data? Wouldn't it be easier to represent using: color (e.g. from blue to red) for sig. increasing/ no-trend / sig. decreasing in observed data, and then hatch the proportion of catchment in which the simulated data failed to reproduce the observed trend?
All your points are valid. For consistency we will change the violinplots to boxplots. We will try your suggestion on the visualization of the trend matching. We are further thinking about splitting the current figure into two figures: a) trend magnitudes (boxplots) and b) trend consistency (barplots).

Lines 418-419: I suggest to add a reference to the figure 8 at the end of the sentence.
Good idea! We now include a reference to figure 8 and 9.

Figures 8 and 9: I find the figure S10 more interesting and more straight to the point than the figures 8 and 9 in the manuscript. I would suggest replacing Figures 8 and 9 by Figure S10.
We appreciate your suggestion! While the figures in the supplementary material give the overall behaviour, the figures 8 and 9 give a closer look into two explicit extreme events. In many studies only the overall behaviour is compared and then from this "general" statements on individual events are drawn. Therefore, we here decided to compare individual events that could be discussed in some more detail and give the overall picture in the supplementary material. Your comment is nevertheless useful as we had to critically reflect on the presentation of results again.

Line 469: What are the results discussed here? Figures S11/S12?
No, we refer to Figure 11. We added the reference to the respective figure & panels at the end of the sentence.

**Discussion section**
Overall comment: One of the objectives of the study was to investigate how the datasets perform in complex terrain, with a focus on mountain region. This should be discussed more here, since you have low/high altitude catchments, the comparison of the behavior of the dataset in different categories of catchment could be investigated in more details. Can we use these datasets for catchments in high altitudes? Why? Why not?What are the recommendations? How could we improve the datasets for these regions?
Thank you for comment. All your points are valid. In response to your comment, we chose to include the elevation information in a more explicit manner in the figures. We now include the median biases of the three elevation bins in various figures.

Line 490: I think that the statement that CERRA clearly improves the representation of precipitations metrics is too strong. The analyses show that it improves how the spread is reproduced over the sample of catchment, however there is no evidence that bias is lower at the catchment scale.What about the spatial correlations of the metrics (obs. vs. reanalysis datasets)?

We still think that this information holds true. As we will switch to showing biases instead of the distribution of the metrics this will become more apparent.

Lines 494-495: I suggest to move the sentence about the precipitation assimilation of ERA5 in the method section (or to remove it) as it adds nothing to the discussion here.

We think that this information should remain here. It is indeed not directly relevant for our results, however, in other regions the advantage of data assimilation might be smaller as ERA5 includes data assimilation of precipitation data itself. Hence, this information is required for consistency reasons and limit the transferability of the results to regions without data assimilation in other reanalysis products.

Lines 499-500: Is there a reference supporting the effect of bubble structures in Europe?

We will do another screening of the literature, however, not many studies have used the CERRA dataset yet. Therefore, this will likely remain our own hypothesis.

Line 505: The snowfalls evaluated are not those that were originally provided in the datasets, I think that it should be reminded here that it was re-calculated for this study (Lines 212-213).

We will clarify this.

Lines 525-527: If datasets and observation agree well at larger scales, should we use them (instead of more refined datasets) when studying large catchments? Should we select different dataset based on catchment size?

Not necessarily. Here, we talk about precipitation variability. Indeed, the results by Monteiro and Morin 2023 suggest that on larger scales the datasets agree well, however, we interpret this that all datasets can represent large scale variability, however, as our results suggest not all datasets can represent smaller scale variability. The "good" representation of the large scale variability is to some extend explainable because of the data assimilation that constrains all reanalysis datasets to represent the large scale climate states. However, as we argue here the smaller scale variability is modulated by topographic features, which are resolution dependent.

If we move away from the specific case of "variability" we could argue that if we only study large catchments, then the choice of the dataset is not as relevant, as all information is smoothed anyway. However, this always depends on what we are interested in. However, if we study catchments across a range of sizes, then we would argue that resolution does play a role, as we can see that locally the differences can be large. Further, we would argue that you should try and have one consistent dataset for all catchments and not select individual datasets for each catchment. Further, as we show in the estimated snowfall analysis, the resolution does matter, and we can reach better performance when we move to higher resolution.

Lines 542-544: As stated below (lines 555-556), the isolated effect of statistical and dynamical downscaling can not be assessed with these datasets, since CERRA uses data assimilation. This sentence should be reworded.

We will rephrase this.

Lines 577-580: Bruno et al. (2023) have also clearly shown how trends (in catchment evapotranspiration) can vary depending on time periods. (Figure 1 in Bruno, G., & Duethmann, D. (2024). Increases in water balance-derived catchment evapotranspiration in Germany during 1970s–

2000s turning into decreases over the last two decades, despite uncertainties. Geophysical Research Letters, 51, e2023GL107753. https://doi.org/10.1029/2023GL107753).

Thank you for the suggested reference. We will check the reference and will decide whether it fits.

Line 597: Errors of > 3 °C during winter could also explain why the snowfall metrics are not well reproduced by some datasets.

Yes. This is exactly what we argue in the following sentence. We could make this more explicit to snowfall here. For example, we could mention snowfall as an example.

**Conclusion**

What the word "variability" (Lines 615, 616) refers to is not totally clear to me: is it temporal variability? spatial variability? Both?

This is temporal variability. We will clarify this.

I think that the summary of the results could be shortened, in the favor of a focus on the strength and weaknesses of the datasets in mountainous areas (which is a gap of knowledge identified Lines 96-97). More recommendations on which dataset to use, where, and why (or why not) would be appreciated.

Thank you for the suggestion. We can certainly extend the recommendations aspect more.

The last sentence is vague and could be removed since all datasets have limitations (CERRA included).

Yes, this is true. We will think about either rephrasing or deleting the sentence.

Figure 12: Could this figure be introduced in the discussion section? It is very useful, although the attribution of performance grade (poorly/satisfactory/well) seems very subjective. Perhaps you could explain how these performance grades were determined in the section on methods?

Indeed, the performance grading is subjective and relies on expert judgment. We will try to make this more transparent or maybe rephrase "performs well/poor/satisfactory" to "limitations apply yes/no/partly". The rephrasing might fit better to the limitations part of the figure. Moving the figure to the discussion section might be a good idea.

**Technical corrections**

Thank you very much for your eye for details. We will directly incorporate the following comments and only reply directly to the ones where we disagree.

Line 65: the name of the product and its spatial resolution are in different orders in this sentence: CERRA at 5.5 km, vs. 6 km COSMO-REA6. I would suggest to rephrase for consistency.
Line 104: "…the representation of the all of these components…"
Line 133 and table 1: "…the Integrated Forecasting System Cy41r2 and covers the period from 1940 until the present". In the text, ERA-5 started in 1940, but in table 1 it started in 1950.
Line 208, Table 2, Figure 3: The metric for annual min and max of temperature are written as "tg_min" and "tg_max" in the text and figure 3, vs. "tg-min" and "tg-max" in table2.
Figure 3: Labels of the metrics: rx1day in the figure vs. Rx1day in table 2 (same thing for r99ptot).
Line 396: "Figure33g" -> Figure 3g
Line 486: "…by CERRA and and overestimated…"
Line 518: "(Bandhauer et al., 2021))". (there is an extra closing parenthesis)
Line 581: "or artificial trends Monteiro and Morin (2023)." I guess that the format of the citation is wrong here.
Line 634: "The CHELSA datasest dataset is available…"

Line 106: "To shed light on the question of which reanalysis products are most suitable…"
We don´t think that including "of" after "question" is necessary in this context.
Line 192: "To calculate catchment averages metrics, we use two complementary…".
We might think of a different formulation.

---

## Author Response (AR1)

**Abstract**

**RC1**: Line 10: I suggest to rephrase the sentence about drought and floods since the datasets do not properly simulate flood and drought events, but rather the conditions (precipitation and temperature) that lead to droughts and floods.

**RC2**: In the abstract, I suggest replacing the CERRA-Land with CERRA since you use both CERRA temperature and CERRA-Land precipitation. The 'drought events' should be specific as 'meteorological droughts', and 'floods' could be replaced by 'extreme precipitation events' or 'heavy precipitation events' as you used in the main text since they are both based on precipitation only.

We have followed the reviewer suggestions, and adapted the abstract accordingly:
Now using *"CERRA"* throughout the abstract.
Specified the extreme event types as *"[...] meteorological droughts"* and *" [...] heavy precipitation events that triggered flooding"*, in the text.

**Introduction**

**RC1**: Lines 28-29: Could you provide a reference for the statement "Among these, the highest quality datasets are those relying on spatially interpolated observations."?

We rephrased the sentence and included references.

> *"Among these, gauge-based and interpolated observations are often seen as 'ground truth' or benchmark (Sun et al., 2018) and still offer the best source of meteorological data for hydrological modelling (Tarek et al., 2020), especially in regions with dense station networks and long homogeneous records."*

Lines 65-66: "many applications require even higher resolution data". Could you mention a few examples and explain why higher resolutions is needed?

We provide additional references in the locations that you suggest.

**RC1:** Line 76: I would suggest specifying the type of drought here (I guess hydrological), since one could also be interested in understanding the drivers of atmospheric drought.

We clarified this; it now says *"hydrological drought"*

**RC1**: Lines 83-91: This paragraph is difficult to follow, as we move from one study to another with different objectives and spatial scales. Also, the conclusions of these studies are missing. We understand that they focused on temporal variations in precipitations (including snow) and their impacts streamflow events, but it could be more explicit.

We extended the description of the references.

> *"For example, Bandhauer et al. (2021) have evaluated ERA5 for precipitation in three mountain regions in Europe, and Lavers et al. (2022) for extreme precipitation for 5,637 precipitation stations quasi-globally. Both studies concluded that ERA5 has deficiencies in modeling precipitation characteristics, such as means, wet day frequency and extremes, compared to high-resolution gridded observations and station data. Gebrechorkos et al. (2024) assessed the value of ERA5-Land precipitation for streamflow simulations worldwide concluding that ERA5-Land performs better than other datasets, but that there is not one single global precipitation dataset that performed best in all catchments. Tarek et al. (2020) assessed the value of ERA5 in North American catchments showing that ERA5 leads to improved*

*hydrological performance compared to its predecessor ERA-Interim, but that observations remain the best source of precipitation data for hydrological modelling. McClean et al. (2023) evaluated the capabilities of global reanalysis products for flood risk modelling in river catchments in Northern England showing lower errors when using the higher resolution ERA5-Land compared to coarser resolution reanalysis datasets. Dura et al. (2024) analyzed seven gridded precipitation products, among them ERA5-Land and CERRA-Land, for their suitability to estimate precipitation enhancement with altitude in France. They found that ERA5-Land underestimates annual precipitation gradients for mid-range mountains and even more in high-altitude regions, while CERRA-Land strongly correlates with annual observed precipitation, but is slightly biased in some regions, and the bias may change the sign according to elevation. Monteiro and Morin (2023) compared ERA5, ERA5-Land and CERRA-Land, among other datasets, concluding that CERRA-Land provides better performance in terms of snow depth and snow seasonality in the European Alps, than ERA5 and ERA5-Land."*

**Data and methods**

**RC1**: Overall comment: the criteria for selecting the four datasets are missing. Why did you select these four datasets? As you mentioned previously, Dura et al (2024) compared seven datasets, why not including all of them in your study?

We now include a justification of our dataset selection.

> *"We selected the newest generation of reanalysis products at high resolution. Hence, the choice of the ERA5 suite (i.e. ERA5 and ERA5-Land) and two of its derivatives (CERRA and CHELSA). The datasets CERRA and CHELSA were chosen as they are expected to be extended back in time offering a valuable tool soon. We chose to exclude other widely used global reanalysis products (i.e., MERRA2 or JRA-55) as they are only available at very coarse resolution (>50 km)."*

**RC1:** Line 186: In my opinion, the first analyses focus more on comparing the spread of the climate metrics (figures 2 and 3), than on the absolute or relative differences between the metrics, which are presented in the supplementary material (more about that in the comments of the results section).

Figures 2 and 3 now show absolute or relative differences between the metrics. See our detailed figure changes below. No changes to text have been made.

***Comments regarding the calculation of metrics:***
**RC1:** Lines 192-196: I didn't find this comparison between time series-based and metric-based approaches in the results. Did you observe differences between these approaches? Anyway, I would suggest to stick to the metric-based approach in the manuscript.
**RC2:** Lines 192-196: Please clarify which way is used to calculate the results presented in the manuscript. I didn't find any comparison between these two methods in the following sections. If both of them are used in different sections to calculate different metrics, they should be mentioned in the corresponding parts.
**RC1:** Line 198 and 204: Are the annual mean and other metrics such as wetdays computed over hydrological years?

We added multiple clarifications throughout the descriptions in the section 3.1.

> *"[…] (i.e. time series-based approach)"*
> *"[…] based on the metric-based approach"*
> *"[…]. All above metrics are based on a calendar year (Jan-Dec)."*
> *"[…] and following the metric-based approach."*

**RC2:** Lines 256-259: Could you explain in more detail how you calculate the cumulative precipitation deficits and SPI6 here? Which reference period is used to calculate the mean precipitation for precipitation deficit? Did the SPI6 calculate based on the moving 6-month window for the whole series and then you choose the one relevant to the event? In my opinion, the SPI already indicates the cumulative precipitation deficits.

We added clarifications in section 3.3.1.

> "[…] Deficits are differences to the long-term mean of cumulative sums (March-August) in 1986–2020."
> "[…] The reference for the SPI-6 are all 6-month sums (March-August) in the period 1986–2020."

**Comments regarding the choice of displayed metrics:**
**RC1:** Line 203: why keep both wet and dry days metrics? Aren't they 100% correlated? The wetdays is shown in Figure 2 (while drydays isn't), and reversely drydays is shown in Figure 7 (but wetdays isn't). I suggest to stick to one or the other of these metrics.
**RC1:** Table 2: The metrics for the mean air temperature is missing in the table. Also, since you used the colddays, why not including hotdays as well (perhaps with a threshold adapted to your catchment sample)?
**RC2:** Lines 202-208: In this part, you introduced the climate metrics considered in the study. However, I was unable to find results related to cwd, cdd, Rx2d, and R95pTot in the subsequent sections or in the supplementary material. Additionally, I am curious why the annual number of hot days was not considered, while cold days were included. Furthermore, I recommend ensuring consistency in the terminology used throughout the manuscript—specifically for the metrics listed in Table 2. For instance, I noticed both 'Rx1d' and 'rx1day' being used. Finally, it might be better to present the results in a more consistent manner. Instead of presenting different metrics across separate figures, consider retaining only a selected set of metrics and using them consistently across all the figures. For example, in Figure 3, you presented rx1day, r99ptot, and wetdays, while rx5day and drydays were omitted. In Figure 5, Rx1day and Rx5day were included but not discussed in the text. In Figure 7, results for rx1day and drydays were shown this time. I think presenting consistent metrics throughout the figures and results, could enhance the clarity and coherence of the manuscript.

We do not include any new metrics; however, we streamlined the presentation of a set of core metrics and their nomenclature, which are recurrently used in different figures.
The core metrics are:
- annual mean of daily mean precipitation (Pmean): Figures 2, 5, 7
- annual mean of daily mean temperature (Tmean): Figures 2, 6, 7
- maximum annual 1-day and 5-day precipitation sums (Rx1d, Rx5d): Figures 3, 5, 7
- annual maximum / minimum of daily mean temperature (tg_max, tg_min): Figures 3, 6, 7
- annual number of wet days (wetdays) and annual number of cold days (colddays): Figures 3, 7
- Fraction of precipitation due to extremely wet days (R99ptot): Figures 3
- Total accumulated solid precipitation (solidprcptot): Figures 4, 7

We adapted the Table 2, to now show all climate metrics that are discussed in the Figures and the manuscript. Missing entries on Pmean and Tmean are now also included.

**Comments regarding the display and discussion of elevational differences:**
**RC1:** Lines 278-280: "The positive reanalysis biases are particularly evident in catchments at high- and low elevations (>2000 and < 1000 m.a.s.l.)" It seems that you have more catchments in these high and low altitudes, perhaps you could provide a histogram of the catchment altitudes?

Line 305: The differences in metrics are described in function of the altitude of the catchments, perhaps you could define the thresholds for low/mid/high altitude catchments once, and refer to these groups of catchments when describing the results?

**RC2:** Lines 284-285: For CHELSA, I agree, however, it seems that the majority of the basins are underestimated by CERRA at low elevations. Similar problems are in Line 303, and Line 305. I suggest adding a threshold to derive the statements here, for example, the percentage of stations that are over- or underestimated compared to all the stations under a specific elevation.

We followed your suggestions and included the information on elevational differences directly in the figures. See detailed comments on figure changes below. We now show median catchment values for three elevation bins (high: >2000m, mid: 1000-2000m, low: ≤1000m). The three elevation bins are close to equal in the number of catchments in each bin. At appropriate points in the text, we have included information on elevation differences.

Further, we have added one sentence to the section 2.3 (methods).

*"The selected catchments cover the three elevation bins low (≤ 1000m, n=32), mid (1000-- 2000m, n=36), and high (> 2000m, n=29) with a comparable number of catchments in each bin."*

**Results**
**Comments regarding Figures:**
**RC1:** Overall comment: In sections 4.1, 4.2, 5.1, the analyses and the figures compare the distribution of the metrics (the boxplots), including all the catchments. From these results we can not conclude that the reanalysis products reproduce well (or not) the different metrics for each catchment. Therefore, I would suggest to plot the distribution of the bias (between obs. and reanalysis data computed for each catchment) instead. This would also help to estimate which dataset over- or under-estimate the metrics (which is difficult to assess on the figures in the manuscript). Or, even better in my opinion, to show the scatter plot of the metrics 'simulated' vs 'observed', as you did for the SPI (e.g. Figure S10). From these, we could assess the correlation between the reanalysis and the observed metrics, as well as their respective spread.

In addition, I don't understand the different types of representations of the results between: a) the box plots in Figures 2, 3 and 5, b) the scatter plots in Figures 4, 8 and 9, c) and the box + violin plots in Figure 7.

a) and b) aimed at the same objective, which is to compare the datasets, why are the results presented in different ways?

c) shows the distribution of the trends, but why adding violin plot? It is redundant with the boxplot (and previous figures did not display violin plots).

**RC1:** Figure 4: The figure would be easier to read with the name of the products on the plot (ERA5, ERA5Land ...).

**RC1:** Figure 5: Rx1day and Rx5day are shown in the figure 5 but not discussed in the text. Moreover, the figure compares the distribution of the metrics, therefore I suggest to change the legend of the figure accordingly (perhaps to something like "Comparison of the distribution of daily to inter-annual precipitation variability"), the same apply for previous figures 2, 3, 5, 6, 7.

**RC1:** Figure 7: This figure is hard to read. Why adding violin plot on top of boxplot? The labels of plots i-o are too close to plots b-h, and can easily be confused with the x-axis of the other plot. The legend for the significance of the trends feel too complicated. The information of interest are: is the trend significant? Is it increasing or decreasing? And do we observe it in the simulated data? Wouldn't it be easier to represent using: color (e.g. from blue to red) for sig. increasing/ no-trend / sig. decreasing in observed data, and then hatch the proportion of catchment in which the simulated data failed to reproduce the observed trend?

Regarding your (RC1) suggestion **"on using scatterplots instead boxplots to compare dataset differences between reanalyses and observations"**, we reiterate our initial response. This would be informative, however by displaying the results in the style of Figure S10 would increase the number of figures dramatically. Each dataset would require its own panel (i.e., as in Figure S10), which would mean that if we want to retain all seasons in figure 2, we will require 40 panels, and 24 panels for figure 3. **Therefore, we will keep the boxplots, but agree with showing the distribution of biases instead of climatological means.**

Lastly, regarding your comment (RC1) on **"our choice to display contents in different ways (Boxplots vs. scatter plots)"**, we reiterate our initial response. **In Figures 2, 3, 5 and 6** we need a concise way to compare multiple metrics in one figure. **In Figures 4, 8 and 9**, we compare single metrics, hence we can allow for a more detailed presentation of the results. **In Figure 7**, we need a way to compare (a) the magnitude in trends and (b) the consistency in trends. This information needs different ways of visualization to convey the relevant information. **Therefore, we choose to keep the general figure styles unchanged.**

Following your suggestions here and in related other comments, **we adapted Figures 2, 3, 4, 5, 6, and 7** to now show dataset differences compared to reference dataset, and we included information on elevation differences in the figures.

See detailed changes for figures 2-7 below:

**Figures 2:**
Figure 2 now displays relative and absolute differences between the catchment means of the reanalysis datasets and observations. In addition, we overlaid the boxplots with markers showing the median biases in the three elevation bins (high: >2000m, mid: 1000-2000m, low: ≤1000m).

The figure caption was updated:

> *"Figure 2. Differences in mean daily precipitation and temperature at the annual and seasonal scale between the catchment means of the reanalysis datasets and observations. Boxplots show differences at the catchment level of (a) precipitation (relative in %) and (b) temperature (absolute in °C) for the entire year (all) and the four meteorological seasons (winter: Dec–Feb, spring: Mar–May, summer: Jun–Aug, fall: Sep–Nov). Boxplots are overlaid with median catchment biases by elevation bin (high: > 2000 m (triangle), mid: 1000–2000 m (square), low: ≤ 1000 m (circle)). Grey shading indicates biases of +/- 10 % (a) and +/- 1 °C (b). Whiskers of the boxplots show min/max."*

The associated text in the manuscript was adapted to the new figures. General statements remain unchanged from the former manuscript version.

> *"[…] however, on the catchment level biases exist with respect to the observations (Figure 2). […] -except for CERRA showing median biases around 0%- both at an annual and seasonal time scale (Figure 2a). In general, precipitation biases don't show a clear elevation dependence, but can show slightly higher biases in lower and higher elevated catchments (see black markers in Figure 2a). […] The positive reanalysis biases are more pronounced in catchments at high- and low-elevations […] (Figure 2a and Figure S1).*
> *In contrast, simulated mean daily temperature is generally slightly underestimated by most reanalysis datasets or matches observations well and most datasets show comparable biases, except for ERA5 (Figure 2b). ERA5 generally has a warm bias –except for winter– while the other reanalyses have a slight cold bias of less than 1 °C (median bias). In winter, all reanalyses show clear cold biases with median catchment biases of at least -1°C and have a strong elevation dependence with larger biases at higher elevations (see Figure 2b (black*

*markers)). ERA5 and CERRA show smaller winter biases (median) than the other datasets. In summer, ERA5_Land, CERRA and CHELSA show small biases (median biases around 0 °C) across most catchments, while ERA5 shows median biases of more than 1 °C with a clear elevation dependence (i.e. warmer biases with higher elevation). In the other seasons (spring and fall) and annually, ERA5 shows warm biases without a clear elevation dependence and spatial pattern (Figure S2), while the other datasets show comparably small cold biases and partially larger biases at higher elevations."*

**Figures 3:**
Same changes, as for Figure 2, apply. The figure now shows relative and absolute differences for different precipitation and temperature metrics.

The figure caption was updated:

*"Figure 3. Differences in selected precipitation (a–d) and temperature (e–g) metrics between the catchment means of the reanalysis datasets and observations. Boxplots show differences at the catchment level for: (a) Rx1d (relative in %), (b) Rx5d (relative in %), (c) R99pTot (absolute in %pts), (d) wetdays (absolute in days), (e) tg_max (absolute in °C), (f) tg_min (absolute in °C), and (g) colddays (absolute in days). Boxplots are overlaid with median catchment biases by elevation bin (high: > 2000 m (triangle), mid: 1000–2000 m (square), low: ≤ 1000 m (circle)). Whiskers of the boxplots show min/max."*

The associated text in the manuscript was adapted in several location to reflect the new figures. General statements remain unchanged from the former manuscript version.

*"[…], except by CERRA showing lower biases, […]. Mean annual maximum daily temperature is slightly overestimated by ERA5 with larger biases at high elevations, and is underestimated by the other datasets. CERRA shows larger biases at higher elevations, while the biases of the other two datasets are less related to elevation (Figure 3 and S6). ERA5 and ERA5-Land show a large spread of positive (low elevation) and negative (high elevation) biases in mean annual minimum daily temperatures. CERRA and CHELSA generally underestimate tg_min, especially at high elevations (Figure 3f, S7). The number of cold days is underestimated by ERA5 and overestimated by ERA5-Land, CERRA and CHELSA, with generally larger biases in high-elevation catchments (Figure 3g, S8)."*

**Figure 4**:
Panels (a-d) now include the name of the reanalysis datasets. No changes made to the figure caption or the text.

**Figure 5**:
Figure 5 now displays relative differences between the four reanalyses and observations for precipitation variability on different timescales. In addition, we overlaid the boxplots with markers showing the median biases in the three elevation bins. The quantification of variability was also adapted to a consistent representation of variability for temperature and precipitation, i.e. standard deviation of anomalies. The method section was adapted to reflect the changes. See comment below on "representation of variability".

The figure caption was updated:

*"Figure 5. Relative differences […] and the extreme metrics (h) Rx1d and (i) Rx5d. The different reanalysis datasets are indicated by the colors. Boxplots are overlaid with median catchment biases by elevation bin (high: >2000 m (triangle), mid: 1000-2000 m (square), low: ≤ 1000 m (circle)). Individual panels can show different scaling of the y-axis. Grey shading indicates biases of +/- 10 %."*

The associated text in the manuscript was adapted in several location to reflect the new figures. General statements remain unchanged from the former manuscript version.

*"[...]. Daily variability is underestimated by all reanalysis datasets with higher biases at higher elevations (Figure 5). For monthly variability, CERRA shows very similar median variability across catchments compared to observations (Figure 5b), but slightly larger biases at higher elevations. The other datasets clearly underestimate variability and also show higher biases at higher elevations.*

*At the inter-annual timescale (Figure 5c), ERA5, ERA5-Land and CHELSA continue to underestimate precipitation variability –without any elevation dependence– and CERRA is again closest to observations (i.e. median biases). However, CERRA overestimates variability in many catchments. If we look at the year-to-year variability in the different seasons (Figure 5d-g), then the general picture prevails that ERA5, ERA5-Land and CHELSA underestimate variability while CERRA matches observed variability well. CERRA shows median biases around zero in winter and spring, and only slightly underestimates variability in summer and fall. Only in fall CERRA shows an elevation dependent underestimation. The other reanalyses show a pronounced elevation dependent underestimation in winter and summer. Surprisingly, the inter-annual variability of the wettest day of the year (i.e., Rx1d) is better represented by ERA5, ERA5-Land, and CHELSA (Figure 5h) than annual and seasonal variability, especially because these datasets clearly underestimate the magnitude of Rx1d (see Figure 3a). CERRA is again closer to observations, but also slightly underestimates variability. We can see similar results for the variability of the maximum 5-day precipitation (Rx5d, Figure 5i)."*

**Figure 6**:
Same changes, as for Figure 5, apply. The figure now shows absolute differences for temperature variability.

The figure caption was updated:

*"Figure 6. Absolute differences in [...] and the extreme metrics (h) tg_max and (i) tg_min. The different reanalysis datasets are indicated by the colors. Boxplots are overlaid with median catchment biases by elevation bin (high: >2000 m (triangle), mid: 1000-2000 m (square), low: ≤ 1000 m (circle)). Individual panels can show different scaling of the y-axis. Grey shading indicates biases of +/- 0.1 °C"*

The associated text in the manuscript was adapted in several location to reflect the new figures. General statements remain unchanged from the former manuscript version.

*"[...] temperature variability (Figure 6), especially on inter-annual timescales. ERA5, CERRA and CHELSA show comparable biases, while ERA5-Land often shows larger negative biases. On daily to monthly time scales, all datasets show an underestimation of variability compared to observations (Figure 6a–b). The inter-annual temperature variability is well represented by all datasets and only ERA5 and CHELSA show a slight overestimation of variability (Figure 6c). All reanalysis datasets agree with observations on the seasonal course of variability and show considerably lower biases –except for winter [...] In spring, [...] (median biases around 0°C). [...] all datasets are closer to observations on the warmest day (tg_max, Figure 6h) than on the coldest day of the year (tg_min, Figure 6i), whose temperature is underestimated by all datasets. For the variability of the coldest day, all datasets show increasing biases with elevation and ERA5, ERA5-Land, and CHELSA show a large spread of biases across catchments. [...] all datasets are more consistent with observations in terms of temperature variability and no single datasets is clearly better than the others. [...] All datasets show a better agreement for the variability of the warmest day of the year (tg_max) than for the coldest day (tg_min), compared to observations."*

**Figure 7:**

We changed the violinplots to boxplots; the boxplots now also contain information on trends by elevation. Further, following your suggestion we simplified the barplots. The barplots now show the percentage of catchments in the observations with a significant positive, no trend, and significant negative trend. This information is overlaid with the percentage of catchments in the reanalysis that agree on the sign&significance or no trend (no hatches), and no agreement between reanalysis and observations (hatches).

The figure caption was updated as follows:

*"Figure 7. Comparison of trend magnitudes and trend significance in precipitation and temperature metrics. (a,e) mean daily precipitation (Pmean; mm), (b,f) maximum 1-day precipitation (Rx1d; mm), (c,g) mean annual amount of solid precipitation (solidprcptot; mm), (d,h) annual number of wet days (wetdays; days), (i,m) mean daily temperature (Tmean; °C), (j,n) annual maximum of daily mean temperature (tg_max; °C), (k,o) annual minimum of daily mean temperature (tg_min; °C), and (l-p) number of cold days (colddays; days). Boxplots show trend magnitudes (theil-sen slopes; 1986-2020) in observations (grey) and reanalysis datasets (colors) across all catchments. Boxplots are overlaid with median trend slopes for the three elevation bins (high: >2000 m (triangle), mid: 1000-2000 m (square), low: ≤1000 m (circle)). The barplots show the percentage of catchments in observations with significant negative trends (blue), no significant trends (grey), and significant positive trends (red) for the respective metric. Each stacked bar is repeated for the reanalysis datasets and is overlaid with the percentage of catchments matching (no hatches) or not matching (hatches) the observed trend in sign and significance. The number above the bar indicates the percentage of catchments where the reanalysis and the observations agree on the sign and significance of trends."*

Only minor modifications to the text were implemented; the statements remain from the former manuscript version.

*"[...] (see Figure 7e, blue)"*
*"[...] (see Figure 7e, non hatched grey bar) "*
*"[...] which leads to an overall lower percentage of catchments where CERRA and observations agree on the sign and significance of trends (see Figure 7e, number above the bar)."*
*"[...] with significant positive trends (opposed to observations), but largely agree on the catchments with no trend"*
*"[...] but also with correct significant decreasing trends."*
*"[...] and the elevation pattern of these trends (Figure 7c, markers"*
*"[...] wet days"*
*"[...] decrease in the number of wet days with larger decreases at lower elevations."*
*"[...] and trend magnitude"*

**Comment on representation of variability:**

**RC1:** Line 337: I suggest to delete the parentheses about "variability from one day to the other" since this is not what the coefficient of variation (or the standard deviation) reflects. You could rank the daily precipitation from the highest to lowest precipitations during a year (which would result in low day to day variation) and get the same standard deviation value.

We removed the statement *"variability from one day to the other"* from the sentence. Further, we adapted our representation of variability to be more consistent between precipitation and temperature. The description of the variability quantification has been adapted in the methods section; Variability is now defined as:

*"To compare the temporal consistency, we analyze the representation of daily, monthly, seasonal, and inter-annual temperature and precipitation variability. We define variability as the standard deviation of daily, monthly, and annual anomalies. Anomalies (absolute anomalies (K) for temperature, relative anomalies (%) for precipitation) are computed using the 1986—2020 climatology of each dataset. Daily anomalies are calculated for each day of the year with a 30-day window centered on the day of interest for the climatology quantification. For monthly anomalies, we first calculate monthly means and then calculate anomalies based on the monthly climatology. Similarly, seasonal anomalies are calculated for winter (DJF), spring (MAM), summer (JJA), and fall (SON). Annual anomalies are based on annual means or extreme metrics (i.e., Rx1d, Rx5d, tg_max, tg_min) and their respective annual climatology. To compare dataset differences, we calculate relative (%, precipitation metrics) and absolute differences (K, temperature metrics) between the variability of the reanalysis and gridded observations for each catchment, respectively."*

**Discussion and conclusion section**
*Comments regarding the discussion:*
**RC1:** Overall comment: One of the objectives of the study was to investigate how the datasets perform in complex terrain, with a focus on mountain region. This should be discussed more here, since you have low/high altitude catchments, the comparison of the behavior of the dataset in different categories of catchment could be investigated in more details. Can we use these datasets for catchments in high altitudes? Why? Why not? What are the recommendations? How could we improve the datasets for these regions?
In response to your comment, we included the elevation information in a more explicit manner in the figures (see detailed comment on figures above). Further, we have discussed elevation differences throughout the manuscript, added the performance evaluation in the newly designed "Summary figure", and we added a new short subsection in the discussion on the *"Performance in high versus low elevation catchments"*.

*"Since we have low to high elevated catchments, we can compare whether the performance of the reanalyses is dependent on elevation. Generally, all reanalysis datasets show comparable performance in low, mid and high elevation catchments. However, for a few metrics and datasets, we can see larger biases in high or low elevated catchments (Figure 12). Especially, temperature metrics related to the cold season show larger biases in the high elevation catchments. For example, in highelevated catchments mean daily winter temperature is colder in all datasets compared to observations (Figure 2, 12), especially CERRA and CHELSA show clear elevation dependent biases. The coldest day of the year (tg_min) is colder in high elevation catchments and in all datasets compared to observations (Figure 3, S7). Also, the number of cold days is overestimated by CERRA and CHELSA at high elevations (Figure 3, S8). Otherwise, we find larger biases in higher elevation catchments for mean precipitation and annual maximum precipitation; especially for ERA5, ERA5-Land and CHELSA (Figure 12)"*

**RC1**: Line 490: I think that the statement that CERRA clearly improves the representation of precipitations metrics is too strong. The analyses show that it improves how the spread is reproduced over the sample of catchment, however there is no evidence that bias is lower at the catchment scale. What about the spatial correlations of the metrics (obs. vs. reanalysis datasets)?
We still think that this information holds true. As we switched to showing biases instead of the distribution of the metrics this statement is now more apparent. **We changed the display of results, but did not change the discussion**, as we see our statement supported by the results.

**RC1**: Lines 542-544: As stated below (lines 555-556), the isolated effect of statistical and dynamical downscaling can not be assessed with these datasets, since CERRA uses data assimilation. This sentence should be reworded.

We changed the sub-section header to *"Differences due to varying modeling techniques"*. Further, we rephrased the suggested sentences; it now reads as follows:

*"As all reanalysis datasets depend to varying degrees on the ERA5 reanalysis but use different modeling techniques to enhance the spatial representation and accuracy of ERA5, we can hypothesize whether these modeling choices may lead to a better performance. CHELSA and ERA5-Land use statistical downscaling with varying complexity to enhance the spatial resolution of ERA5, while CERRA uses a combination of dynamical downscaling and additional data assimilation to enhance spatial resolution and the representation of precipitation. Statistical downscaling did not substantially improve the representation of temperature and precipitation metrics compared to ERA5 and observations (Figures 2, 3). [...]"*

***Comments regarding the conclusion***
RC1: What the word "variability" (Lines 615, 616) refers to is not totally clear to me: is it temporal variability? spatial variability? Both?
We added "*temporal*" to clarify this.

RC1: I think that the summary of the results could be shortened, in the favor of a focus on the strength and weaknesses of the datasets in mountainous areas (which is a gap of knowledge identified Lines 96-97). More recommendations on which dataset to use, where, and why (or why not) would be appreciated. The last sentence is vague and could be removed since all datasets have limitations (CERRA included).
Regarding your comment on **"weaknesses of the datasets in mountainous areas"** we have included a short new section *"Performance in high versus low elevation catchments"* in the discussion section (see comment above on this topic).
In addition, we have shortened the summary part, extended the recommendation aspect and removed the last sentence. We have adapted the text as follows:

*"In this paper, we [...]. Across the various precipitation and temperature metrics, and their temporal variability, the CERRA dataset best represents the observations (Figure 12, S17), making it a good overall choice for hydrological impact studies in low to high elevated catchments. In the following, we will give a few general recommendations for the use of reanalysis datasets compared and some more specific recommendations for some hypothetical use-cases.*

*As precipitation is the dominant variable to drive the hydrological response, we can recommend using the CERRA dataset for hydrological impacts studies and can hypothesize that CERRA will likely serve as a good input for hydrological models, because: (a) CERRA captures mean and extreme precipitation well across seasons and elevation; (b) captures the three heavy precipitation events --which triggered flooding in Switzerland-- well in terms of intensity and spatial extent; and (c) can in combination with its good representation of temperature lead to small biases in snowfall fraction and total snow amount. In snow dominated catchments we can further recommend the use of CHELSA, as it also represents snow fraction and amount well.*

*Many climate change impact studies require bias-adjustment of climate model simulations prior to modeling hydrological changes. Here, we can recommend using CERRA as the reference for the bias-adjustment, for similar reasons as above and because of its good representation of wet-day frequency. The use of ERA5, ERA5-Land or CHELSA will likely be an*

*insufficient reference to correct for biases, as these datasets themselves are biased in wet-day frequency, and mean and extreme precipitation.*

*Overall, all four reanalysis datasets can represent extreme dry and wet meteorological conditions of varying severity and intensity as illustrated for the droughts in 2003 and 2018 and the extreme precipitation events in 1999 and 2005. For studying meteorological droughts in mountain regions, we recommend using ERA5, ERA5-Land and CHELSA. While they can overestimate drought severity, they can overall capture these events well. CERRA on the other hand underestimates the severity and intensity of the studied drought events, revealing spurious precipitation events that alleviate drought conditions, which seems to be the major limitation of this dataset. For heavy precipitation events we recommend using CERRA. CERRA best represents wet extremes, while the other datasets overestimate spatial extents, severity and intensity in catchments with low to moderate precipitation characteristics.*

*In conclusion, [...].”*

Figure 12: Could this figure be introduced in the discussion section? It is very useful, although the attribution of performance grade (poorly/satisfactory/well) seems very subjective. Perhaps you could explain how these performance grades were determined in the section on methods? We have adapted the summary figure to be more transparent and extended the quality assessment to now also include the performance in the three elevation bins. Further, we have introduced Figure 12 in the discussion, by including multiple figure references in the discussion part. The new summary figure 12 is a simplification of the more detailed figures S14-16 in the supplementary material summarizing the biases of all metrics. In Figure 12 the performance is color coded into three categories of overestimation (orange colors) and underestimation (purple colors). The color code represents the biases shown in Figure S14-16 in the supplementary figure. In the supplementary material we give some more detail on the bias quantification. But in general, it is a summary of the biases shown in the manuscript figures.

*“In the following figures we have summarized all biases from the four reanalysis datasets (ERA5, ERA5-Land, CERRA, CHELSA) compared to the gridded observations. The biases are reflective of the results shown in the main manuscript. For each mean, extreme and temporal variability metric of precipitation/temperature the median bias (absolute or relative) in high (>2000 m), mid (1000-2000 m) and low ≤1000 m) elevation catchments, and all catchments is given. The biases in metrics are divided into three categories of positive and negative biases; the respective bounds of bias categories are given in the figure legends. For the two extreme event types -meteorological drought (droughts) and 2-day heavy precipitation events (floods)- we quantify the three performance metrics: detection, intensity and extent. The detection is representative of the percentage of catchments in the observations that are also correctly detected in the reanalysis dataset. That means the optimal detection is 100% of catchments in the observations also trigger a detection (drought: SPI6<-1, floods: SPI-2day > +1.5) in the reanalysis. We subtract the percentage value by 100% (optimal value) to receive the underestimation in detection in percent. The intensity gives the relative bias in drought deficit (drought) and 2-day precipitation sum (floods) as the median bias for the subset of catchments that show an extreme event in the observations. The bias in extent is calculated as the ratio of the number of catchments in the reanalysis with event detection divided by the number of catchments in the observations with event detection minus 1; multiplied by 100 to get percentages. This gives in percentage the over-/underestimation of the drought/flood extent. For the biases in trends, we determine the correct detection of significant (i.e. agreement in sign and significance) and non-significant observed trends separately. That means the optimal detection is 100\% of catchments in the observations also trigger a detection of (non-)significant trends in the reanalysis. We subtract the percentage value by 100% (optimal value) to receive the underestimation in detection in percent. We evaluate the*

*agreement in trends in high (>2000 m), mid (1000-2000 m), and low (≤1000 m) elevation catchments, and all catchments separately. Further, we evaluate the bias in the spatial extent of significant trends defined as the ratio of the number of catchments in the reanalysis with significant trends divided by the number of catchments in the observations with significant trends minus 1; multiplied by 100 to get percentages. For both the trend agreement and the extent, we exclude all cases where less than 10 catchments in the observations show a significant trend.".*

**Miscellaneous minor comments**

**RC2:** Line 264: I think it should be August 2005 here. Similar to drought, why do you use both 2-day sums and 2-day SPI? From the parts of the result, I also did not see significant differences between these two ways.

We corrected this; now it says *"August"*.

**RC2:** Lines 294-296: I am not sure if the number of wet days will influence the annual maximum 1-day accumulated precipitation.

We rephrased this; it now says:

*"Rx1d, Rx5d, and r99ptot are underestimated by ERA5, ERA5-Land, and CHELSA across all elevation zones (Figure 3a-c, S3 and S4) as precipitation is distributed across too many days with moderate precipitation intensity in all of these products, especially in catchments at higher elevations (Figure 3d, S5)."*

**RC2:** Lines 299-301: It is better to add reference figures from supplementary material here.

We modified the text and now include figure references to Figure 3 (now including information on elevation) and to the supplementary material.

**RC2:** Line 381: I think it should be Figure 7b.
Line 396: I think it should be Figure 3g.

The two figure references have been corrected. We checked figure referencing throughout the manuscript and applied changes where necessary.

**RC2:** Lines 466-467, 471-472, 473-474, 477-478, 486-487: I was unable to find results related to the spatial extent for all four reanalysis datasets.

We included the respective information in the supplementary material (added Figures S13 and S16) and added appropriate figure references at the indicated locations.

[Figure]

*Figure S13: Standardized 2-day catchment precipitation sums of the three extreme precipitation events in the observations and reanalysis datasets. (a, left column) shows the standardized values for the first event (11–12 May 1999), (b, center column) for the second (21–22 May 1999), and (c, right column) for the third event (21–22 Aug 2005). Each colored dot represents a single catchment.*

**RC1:** Line 332: Did you mean that CHELSA overestimates solid precipitation? (the world "solid" is missing)

Yes, thank you for noticing. We added *"solid"* for clarification.

**RC1:** Line 349: I suggest to add a reference to the figure 6 and the end of this sentence.

Added the reference to figure 6.

**RC1:** Line 385: On figure 7d, the proportion of catchments with significant trend in Rx1day seems closer to 20% than 10%.

True, we adapted this accordingly.

**RC1:** Lines 418-419: I suggest to add a reference to the figure 8 at the end of the sentence.

We now include a reference to figure 8 and 9.

**RC1:** Line 469: What are the results discussed here? Figures S11/S12?

No, we refer to Figure 11. We added the reference to the respective figure & panels at the end of the sentence. *"(Figure 11e,f,h)"*

**RC1:** Line 505: The snowfalls evaluated are not those that were originally provided in the datasets, I think that it should be reminded here that it was re-calculated for this study (Lines 212-213).

We will clarified this; it now reads:

*"This difference may be related to the scale difference between the two analyses or that they used the snow outputs from the reanalysis directly, while we approximated snow from precipitation and temperature."*

**RC1:** Line 597: Errors of > 3 °C during winter could also explain why the snowfall metrics are not well reproduced by some datasets.

Yes. This is exactly what we argue in the following sentence. We made this more explicit in the text; it now reads as follows:

*"[…], especially in winter (e.g., for the snowfall approximation), […]"*

**RC1:** Lines 577-580: Bruno et al. (2023) have also clearly shown how trends (in catchment evapotranspiration) can vary depending on time periods. (Figure 1 in Bruno, G., & Duethmann, D. (2024). Increases in water balance-derived catchment evapotranspiration in Germany during 1970s–2000s turning into decreases over the last two decades, despite uncertainties. Geophysical Research Letters, 51, e2023GL107753. https://doi.org/10.1029/2023GL107753).

We added the reference.

*"Also Bruno and Duethmann (2024) have shown that the sign and significance of trends can vary depending on the time."*

**RC1:** Line 65: the name of the product and its spatial resolution are in different orders in this sentence: CERRA at 5.5 km, vs. 6 km COSMO-REA6. I would suggest to rephrase for consistency.

We changed the text accordingly.

*"[…] and the COSMO-REA6 from the German Weather Service at 6 km resolution […]"*

**RC1**: Line 104: "...the representation of  all of these components..."

We removed *"the"*.

**RC1:** Line 133 and table 1: "...the Integrated Forecasting System Cy41r2 and covers the period from 1940 until the present". In the text, ERA-5 started in 1940, but in table 1 it started in 1950.

We adapted the information in table 1. It now says *"1940"*

**RC1:** Line 208, Table 2, Figure 3: The metric for annual min and max of temperature are written as "tg_min" and "tg_max" in the text and figure 3, vs. "tg-min" and "tg-max" in table2.

We adapted the table and checked the manuscript for consistency. It now says "tg_min" / "tg_max" throughout the manuscript, figures, and tables.

**RC1:** Figure 3: Labels of the metrics: rx1day in the figure vs. Rx1day in table 2 (same thing for r99ptot).

We adapted the table, figures and checked the manuscript for consistency. It now says *"Rx1d"* and *"R99ptot"*.

**RC1:** Line 396: "Figure33g" -> Figure 3g

Changed to *"Figure 3d"*

**RC1**: Line 486: "...by CERRA  and overestimated…"

We removed one of the *"and"*.

**RC1**: Line 518: "(Bandhauer et al., 2021))". (there is an extra closing parenthesis)

We removed the parenthesis.

**RC1**: Line 581: "or artificial trends Monteiro and Morin (2023)." I guess that the format of the citation is wrong here.

We changed the referencing style to *"(Monteiro and Morin, 2023)"*

**RC1**: Line 634: "The CHELSA  dataset is available…"

We corrected the spelling mistake, and it says *"dataset"* now.

**RC1**: Line 192: "To calculate catchment averages metrics, we use two complementary…".

We rephrased this.

*"To calculate metrics on the catchment level, we use two complementary …"*

**Comments without any changes to the manuscript:**
**RC1**: Line 182: I wonder why you did not used precipitation and temperature provided by the CAMELS-CH? Do they come from the same products?

The meteorological data in CAMELS-CH are based on the same data from MeteoSwiss as we used in our study. Other available meteorological data within the CAMELS-CH dataset are post-processed MeteoSwiss data from the PREVAH hydrological model. We chose to use the original MeteoSwiss products instead to have full flexibility on the analysis. Especially for the climate indicators, which we calculated on the grid cell first and then averaged over the catchments, required us to use the gridded data and not the timeseries data from CAMELS-CH. No changes to the manuscript were made.

**RC1**: Lines 211-212: How different are the estimations of snowfall (re)computed in your study from those provided by the datasets?

We did not quantify this. We chose to use a consistent approach to separate liquid and solid precipitation as not all datasets provide snow height or snow water equivalent or the separation of liquid and solid precipitation. Further, in many applications (e.g., hydrological models or land surface models) snow is calculated within the model itself often following a similar approach using a temperature index approach. No changes to the manuscript were made.

**RC1**: Lines 312-315 How is this short section about the relationship between the fraction of snow and altitude relevant for the study (Figure 4e)?

To give some context. No changes to the text were made.

**RC1**: Line 319: The 100% of under- and over-estimation could be related to catchment with very low fraction of snow.

Yes, certainly. No changes to the text were made.

**RC1:** Lines 494-495: I suggest to move the sentence about the precipitation assimilation of ERA5 in the method section (or to remove it) as it adds nothing to the discussion here.

We think that this information should remain here. It is indeed not directly relevant for our results, however, in other regions the advantage of data assimilation might be smaller as ERA5 includes data assimilation of precipitation data itself. Hence, this information is required for consistency reasons and limit the transferability of the results to regions without data assimilation in other reanalysis products.

**RC1:** Lines 499-500: Is there a reference supporting the effect of bubble structures in Europe?

We did another screening of the literature, however, not many studies have used the CERRA dataset yet. Therefore, this remains our own hypothesis. No changes to the manuscript.

**RC1:** Lines 525-527: If datasets and observation agree well at larger scales, should we use them (instead of more refined datasets) when studying large catchments? Should we select different dataset based on catchment size?

Not necessarily. Here, we talk about precipitation variability. Indeed, the results by Monteiro and Morin 2023 suggest that on larger scales the datasets agree well, however, we interpret this that all datasets can represent large scale variability, however, as our results suggest not all datasets can represent smaller scale variability. The "good" representation of the large scale variability is to some extend explainable because of the data assimilation that constrains all reanalysis datasets to represent the large scale climate states. However, as we argue here the smaller scale variability is modulated by topographic features, which are resolution dependent.

If we move away from the specific case of "variability" we could argue that if we only study large catchments, then the choice of the dataset is not as relevant, as all information is smoothed anyway. However, this always depends on what we are interested in. However, if we study catchments across a range of sizes, then we would argue that resolution does play a role, as we can see that locally the differences can be large. Further, we would argue that you should try and have one consistent dataset for all catchments and not select individual datasets for each catchment. Further, as we show in the estimated snowfall analysis, the resolution does matter, and we can reach better performance when we move to higher resolution.

**RC1:** Line 106: "To shed light on the question of which reanalysis products are most suitable..."

We don´t think that including "of" after "question" is necessary in this context.

**RC1:** Figures 8 and 9: I find the figure S10 more interesting and more straight to the point than the figures 8 and 9 in the manuscript. I would suggest replacing Figures 8 and 9 by Figure S10.

We appreciate your suggestion! While the figures in the supplementary material give the overall behaviour, the figures 8 and 9 give a closer look into two explicit extreme events. In many studies only the overall behaviour is compared and then from this "general" statements on individual events are drawn. Therefore, we here decided to compare individual events that could be discussed in some more detail and give the overall picture in the supplementary material. No changes to the figures have been made.

**RC1:** Line 394-396: The reasons (poor representation of dry days) for the poor trends detection in those metrics should be moved in the discussion section.

We decided to keep this statement in the results part. No changes applied.